# Dithranol targets keratinocytes, their crosstalk with neutrophils and inhibits the IL-36 inflammatory loop in psoriasis

Theresa Benezeder[1†], Clemens Painsi[2†], VijayKumar Patra[1], Saptaswa Dey[1], Martin Holcmann[3], Bernhard Lange-Asschenfeldt[2], Maria Sibilia[3], Peter Wolf[1]*

[1]Department of Dermatology, Medical University of Graz, Graz, Austria; [2]State Hospital Klagenfurt, Klagenfurt am Wörthersee, Austria; [3]Institute of Cancer Research, Department of Medicine I, Comprehensive Cancer Center, Medical University of Vienna, Vienna, Austria

**Abstract** Despite the introduction of biologics, topical dithranol (anthralin) has remained one of the most effective anti-psoriatic agents. Serial biopsies from human psoriatic lesions and both the c-Jun/JunB and imiquimod psoriasis mouse model allowed us to study the therapeutic mechanism of this drug. Top differentially expressed genes in the early response to dithranol belonged to keratinocyte and epidermal differentiation pathways and IL-1 family members (i.e. *IL36RN*) but not elements of the IL-17/IL-23 axis. In human psoriatic response to dithranol, rapid decrease in expression of keratinocyte differentiation regulators (e.g. involucrin, *SERPINB7* and *SERPINB13*), antimicrobial peptides (e.g. ß-defensins like *DEFB4A, DEFB4B, DEFB103A*, S100 proteins like *S100A7, S100A12*), chemotactic factors for neutrophils (e.g. *CXCL5, CXCL8*) and neutrophilic infiltration was followed with much delay by reduction in T cell infiltration. Targeting keratinocytes rather than immune cells may be an alternative approach in particular for topical anti-psoriatic treatment, an area with high need for new drugs.

*For correspondence:
peter.wolf@medunigraz.at

†These authors contributed equally to this work

Competing interests: The authors declare that no competing interests exist.

## Introduction

With the development and market introduction of biologics, much progress has been made in recent years in the systemic treatment of psoriasis. Currently, targeted therapy with antibodies against IL-17 or IL-23 exhibits high efficacy and allows complete or almost complete clinical clearance of psoriasis lesions in a high percentage of cases (*Blauvelt et al., 2017*; *Langley et al., 2014*; *Papp et al., 2017*; *Reich et al., 2018*). However, patients with limited body area involvement (i.e. mild forms of psoriasis) who represent the majority of psoriasis patients with up to 90% (*Stern et al., 2004*) have been neglected. Not much innovation has occurred in the past few years on topical treatment of psoriasis that is mainly prescribed to these patients. Steroids, vitamin D3 and vitamin A analogues are most commonly used as topical agents in such patients, but besides changes in their pharmaceutical formulation, they have not been developed further in recent years (*de Korte et al., 2008*; *Krueger et al., 2001*; *Langley et al., 2011*; *Schaarschmidt et al., 2015*). Short courses of dithranol (1,8-dihydroxy-9-anthracenone or anthralin) have been successfully used as intermittent topical treatment of psoriasis since 1916 (*Nast et al., 2012*; *Sehgal et al., 2014*; *Vleuten et al., 1996*). Despite its numerous disadvantages like brown staining and irritation of perilesional skin, dithranol has remained one of the most effective topical treatment modalities in psoriasis. Analogous to the most recent generation of biologics (including anti-IL-17 and anti-IL-23 antibodies), dithranol delivers PASI75 rates in 66–82.5% of patients with fast action and clearance of skin lesion within very few weeks (*Kemény et al., 1990*; *Painsi et al., 2015b*; *Sehgal et al., 2014*; *van de Kerkhof, 2015*). Although dithranol has been used for many years, its exact mechanism of action has remained

largely unknown (*Holstein et al., 2017*; *Kemény et al., 1990*; *Kucharekova et al., 2006*; *Sehgal et al., 2014*).

With the aim of unraveling dithranol's therapeutic mechanisms and to possibly uncover new targets for topical treatment of psoriasis, we conducted a clinical trial and employed several mouse models including the c-Jun/JunB knockout model (*Zenz et al., 2005*) and the imiquimod psoriasis model (*van der Fits et al., 2009*) in order to address this issue and elucidate dithranol's effects. In this study, we demonstrate that dithranol exerts its anti-psoriatic effects by directly targeting keratinocytes and their crosstalk with neutrophils, as well as disrupting the IL-36 inflammatory loop. Consistent with this finding, we observed that dithranol's therapeutic activity was completely independent of its pro-inflammatory effect mainly on perilesional skin, thus overthrowing the long-believed paradigm that dithranol-induced irritation is crucial for its anti-psoriatic action and unraveling irritation merely as a bystander effect of treatment.

## Results

### Topical dithranol leads to fast reduction in PASI score linked to decrease in epidermal hyperproliferation and delayed reduction of inflammatory infiltrate in psoriatic skin

As depicted in *Figure 1C* (and *Figure 1—figure supplement 1*), dithranol did lead to a fast reduction of psoriatic skin lesions in most of the 15 patients of the study, as determined by psoriasis area and severity index (PASI) and local psoriasis severity index (PSI) of marker lesions. As shown in *Table 1*, the mean decrease in PASI score was 58% after 2–3 weeks, confirming its high clinical efficacy (*Painsi et al., 2015a*; *Painsi et al., 2015b*; *Swinkels et al., 2004*). Remarkably, dithranol treatment did lead to a visible inflammatory response only at perilesional skin sites, but not within psoriatic plaques and there was no correlation between dithranol-induced erythema and its anti-psoriatic effect (*Figure 1—figure supplement 2*). The clinical response was confirmed by the results of histological analysis of skin biopsies taken throughout the dithranol treatment course (*Figure 1A and B*). Dithranol application led to a significant reduction in epidermal hyperplasia (as measured by thickness of epidermis). Intriguingly, there was no significant change in dermal infiltrate score during treatment. At the follow-up visit, hyperplasia of the epidermis was reduced further and cellular infiltrate in the dermis was significantly diminished (*Figure 2*).

### Clinical response to dithranol in psoriasis patients is linked to I) fast upregulation of keratinization genes and downregulation of neutrophil chemotactic genes and neutrophilic infiltration followed by II) delayed downregulation of inflammatory response-related genes and proteins

Dithranol slowly diminished CD3, CD4, FoxP3 and CD8 cell counts in the dermis as evidenced by unaffected cell numbers early on during the treatment course and significant reduction only at the follow-up visit (4–6 weeks after end of treatment) (*Figure 2*). The response of T cells in the epidermis occurred a little earlier, with significant reduction present already at the end of treatment (week 2–3) (*Figure 2*). In agreement with other studies (*Holstein et al., 2017*; *Swinkels et al., 2002a*; *Vleuten et al., 1996*; *Yamamoto and Nishioka, 2003*), we found a fast decrease in epidermal hyperproliferation (epidermal thickness, Ki-67 and CK16 staining) at end of treatment (week 2–3). Neutrophil numbers (as assessed by myeloperoxidase staining) in epidermis and dermis were also significantly reduced at that time point. An increase in the number of Langerhans cells in the epidermis (as indicated by CD1a staining) was detected at the follow-up visit.

Microarray analysis comparing lesional skin at day 6 of treatment with lesional skin samples at baseline revealed that dithranol led to differential expression of 62 genes including 17 genes with reduced and 45 genes with increased expression. Among these differentially expressed genes (DEGs), there was a significant upregulation of genes involved in keratinization and keratinocyte differentiation (e.g. *KRT2, LCE1C, KRT73, LCE1A*) and establishment of skin barrier (e.g. *FLG2, HRNR, FLG*). Furthermore, dithranol downregulated genes of antimicrobial peptides (AMPs) such as ß-defensin-2 (*DEFB4A, DEFB4B*) and chemoattractants for neutrophils (e.g. *CXCL5, CXCL8, PPBP1, TREM1*) (*Table 2*).

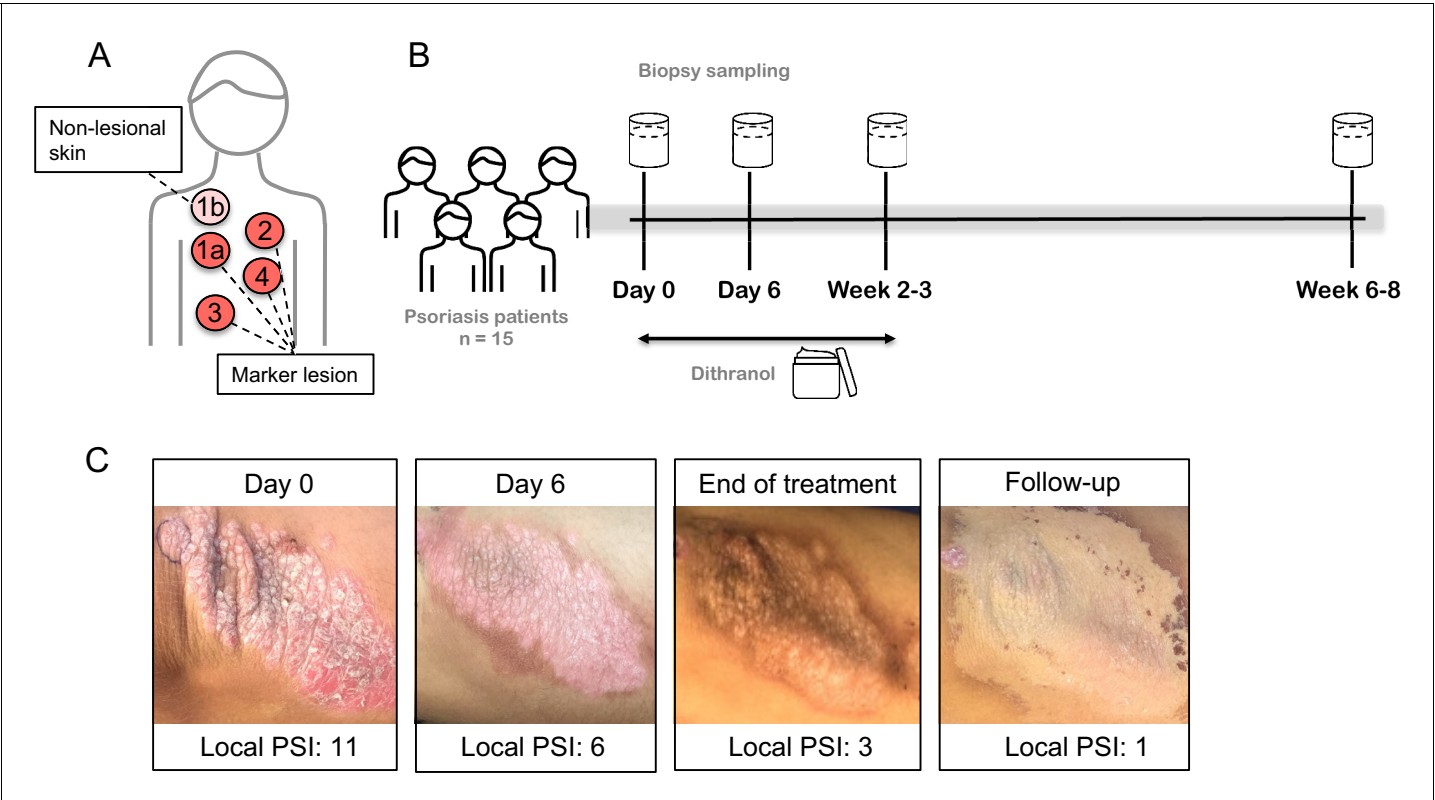

**Figure 1.** Dithranol leads to a fast reduction of psoriatic skin lesions as determined by local psoriasis severity index (PSI) of marker lesions. (A,B) 15 psoriasis patients were treated with dithranol. Skin biopsies were taken from marker lesions at multiple timepoints: 1a = lesional skin at baseline, 2 = lesional skin at day 6, 3 = lesional skin at end of treatment, week 2–3, 4 = lesional skin at follow-up, 4–6 weeks after end of treatment. In addition, non-lesional skin at baseline (1b) was sampled. (C) Representative images of lesional psoriatic skin and local psoriasis severity index (PSI; sum of erythema (0–4), induration (0–4) and scaling (0–4); 0 = none, 1 = mild, 2 = moderate, 3 = severe, 4 = very severe) at different time points.

The online version of this article includes the following source data and figure supplement(s) for figure 1:

**Figure supplement 1.** Psoriasis area and severity (PASI) score at baseline (day 0), early during treatment (day 6), at end of treatment (week 2–3) and at follow-up (4–6 weeks after end of treatment) for all 15 patients (A1–A15) treated with dithranol.

**Figure supplement 1—source data 1.** Values displayed in graph shown in *Figure 1—figure supplement 1*.

**Figure supplement 2.** Lesional and perilesional erythema does not correlate with reduction in PASI score.

At the end of treatment (week 2–3), dithranol had deregulated a total of 453 genes, with downregulation of 325 DEGs and upregulation of 128 genes. Compared to day 6, expression of various genes involved in keratinocyte differentiation decreased further upon dithranol treatment and other AMPs (e.g. *S100A7A, S100A12, DEFB103A*) and genes involved in neutrophil-mediated inflammatory responses significantly diminished in their expression. With delay, dithranol also lowered expression of inflammatory response-related genes (e.g. *IL1B, IL17, IL22, IL36A, IL36G, IL36RN*) only at the end of treatment (week 2–3) (*Table 3*). Notably, genes involved in T-cell activation were not differentially expressed in the observation period (during dithranol up to week 2–3). To verify a subset of differentially expressed genes from the microarray data, we performed nCounter Nanostring analysis on 80 target and four reference genes. Ratios of microarray target genes strongly correlated with those obtained from Nanostring analysis at the early (day 6) (r = 0.8830) and late time point examined at the end of dithranol treatment (week 2–3) (r = 0.8859) (*Supplementary file 2*). Comparing day 6 vs. baseline, we found gene ontology (GO) terms related to keratinization (e.g. keratinocyte differentiation, establishment of skin barrier) and neutrophil chemotaxis (e.g. neutrophil migration, regulation of neutrophil migration) among the most significant GO groups (*Table 4*). Top significantly enriched GO terms at week 2–3 vs. baseline were related to immune response (e.g. inflammatory response, cytokine secretion) and differentiation of keratinocytes (e.g. epidermis development, keratinization) (*Table 5*). At follow-up (4–6 weeks after end of treatment), 10 of 13 patients

**Table 1.** Psoriasis area and severity index (PASI) and psoriasis severity index (PSI) from 15 psoriasis patients.

| Parameter | | Time point | | | |
|---|---|---|---|---|---|
| | | Baseline | Day 6 | End of treatment | Follow-up |
| PASI | | | | | |
| | mean ± SD | 13.6 ± 10.3 | 9.0 ± 6.3 (p<0.0001)[*] | 5.1 ± 3.8 (p=0.0001)[*] | 5.7 ± 6.7 (p=0.0002)[*] |
| | %reduction, mean ± SD (range) | - | 32.9 ± 8.0 (16.7–44.3) | 57.5 ± 9.5 (41.8–74.2) | 56.1 ± 23.3 (3.1–81.0) |
| Local PSI | | | | | |
| | mean ± SD | 6.7 ± 1.1 | 3.3 ± 1.6 (p<0.0001)[*] | 2.0 ± 1.5 (p<0.0001)[*] | 1.6 ± 1.3 (p<0.0001)[*] |
| | %reduction, mean ± SD (range) | - | 52.6 ± 21.9 (14.3–100) | 69.0 ± 23.9 (16.7–100) | 76.3 ± 18.3 (37.5–100) |

[*]P value was determined using Wilcoxon test comparing indicated value to baseline.

The online version of this article includes the following source data for Table 1:

Source data 1. Values displayed in *Table 1*.

showed >50% reduction in epidermal thickness and >75% reduction in Ki67 staining. These 10 patients were classified as histological responders. Microarray gene expression analysis at end of treatment (week 2–3) revealed that 131 genes were differentially expressed in histological responders compared to non-responders, predicting histological outcome for the follow-up time point 4–6 weeks after the end of treatment. Using gene ontology enrichment analysis of these DEGs, we found pathways like keratinocyte differentiation, cornification and keratin filament formation among the top 20 GO terms (*Supplementary file 3*).

## Topical application of dithranol ameliorates psoriasis-like skin lesions in c-Jun/JunB knockout mice by directly targeting keratinocyte genes and inhibiting *IL36RN*

To study the effect of dithranol in a keratinocyte-based psoriatic mouse model, we used c-Jun/JunB knockout mice (treatment protocol shown in *Figure 3A*). Topical dithranol application strongly reduced psoriatic lesions as measured by macroscopic overall ear thickness and microscopic epidermal thickness in this genetic model of psoriasis based on inducible epidermal deletion of the AP1 transcription factors c-Jun/JunB (*Glitzner et al., 2014*; *Zenz et al., 2005*; *Figure 3B–D*). There was no difference in the outcome to dithranol with regard to different concentrations series, therefore, data was pooled for certain analyses as indicated (*Figure 3*). Gene expression profiling using Clariom S mouse microarray showed that among the top 45 differentially regulated genes, dithranol downregulated genes belonging to the group of late cornified envelope genes (*Lce6a, Lce1i, Lce1g, Lce1f*), as well as hornerin (*Hrnr*) (*Table 6*). Gene ontology enrichment analysis of all differentially expressed genes (fold change >1.5 and p-value<0.05) revealed skin development, epidermis development, epithelial cell differentiation and keratinization among the top 20 significantly enriched pathways (*Table 7*). Associated genes found were for example *Flg2, Ivl, Hrnr, Lor, Krt2, Casp14, Lce1c, Serpinb7* and *Serpbinb13*, among others. We then compared DEGs from psoriasis patients treated with dithranol to DEGs from dithranol-treated c-Jun/JunB knockout mice (*Figure 4*). In total, the microarray assay allowed us to compare 300 DEGs in dithranol-treated psoriasis patients (week 2–3 vs. day 0) and 18 DEGs from day 6 vs. day 0 with 336 murine DEGs from dithranol-treated c-Jun/JunB knockout mice. Notably, comparing DEGs from day 6 vs. day 0 with murine DEGs, we found an overlap of 11 genes. Among these genes were *CASP14*, a non-apoptotic caspase involved in epidermal differentiation and *FLG2, HRNR, KRT2, LCE1C* and *SERPINB12*, involved in keratinocyte differentiation and epidermis development (*Bergboer et al., 2011*; *Henry et al., 2011*; *Henry, 2012*; *Kuechle et al., 2001*; *Sandilands et al., 2009*; *Sivaprasad et al., 2015*; *Toulza et al., 2007*). Even more strikingly, the overlap between DEGs from week 2–3 vs. day 0 and murine DEGs comprised 20

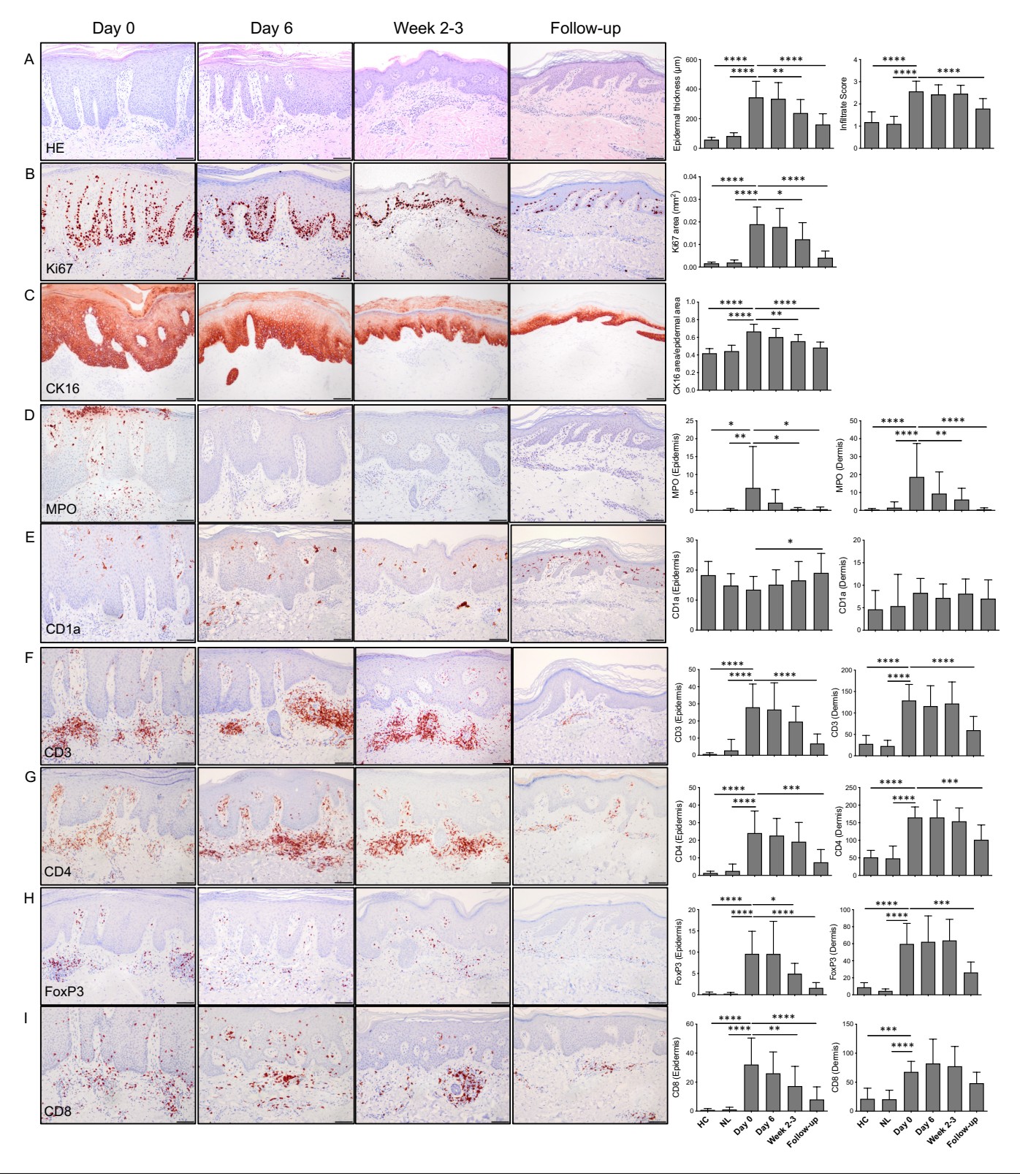

**Figure 2.** Histological and immunohistochemical analysis of lesional skin before (day 0), during (day 6), at end of treatment (week 2–3) and 4–6 weeks after ending treatment (follow-up). (**A**) Representative H and E images, epidermal thickness and cellular infiltrate scoring. In treated psoriatic lesions, epidermal thickness was significantly decreased at week 2–3 and at follow-up. Infiltrate score (0 = none, 0.5 = none/low, 1 = low, 1.5 = low/moderate, 2 = moderate, 2.5 = moderate/high, 3 = high infiltration of immune cells) was significantly higher in untreated psoriatic lesions compared to non-

*Figure 2 continued on next page*

*Figure 2 continued*

lesional skin (NL) and healthy controls (HC). In treated psoriatic lesions, infiltrate score was significantly decreased at follow-up. (B) Ki67 staining in epidermis was significantly reduced at week 2–3 and follow-up. (C) CK16 staining in epidermis was significantly reduced at week 2–3 and follow-up. (D–I) Representative IHC images and mean cell counts for epidermis and dermis. Neutrophil cell counts were significantly reduced at week 2–3 and follow-up in epidermis and dermis (D). Langerhans cell (CD1a+) numbers in epidermis were significantly increased at follow-up (E). Epidermal FoxP3 (H) and CD8 (I) cell counts were significantly reduced at week 2–3. Dermal CD3 (F), CD4 (G), and FoxP3 (H) positive cell counts display significant reduction at follow-up. One-way ANOVA with Dunnet's multiple comparisons test was used for statistics. Bars represent mean ± SD; *p≤0.05; **p≤0.01; ***p≤0.001; ****p≤0.0001; scale bar = 100 μm.

The online version of this article includes the following source data for figure 2:

**Source data 1.** Values displayed in bar plots shown in *Figure 2*.

---

genes including *IL36RN, IVL, SERPINB7* and *SERPINB13*, that were all increased in lesional skin at baseline compared to non-lesional skin and downregulated by dithranol treatment. Findings from microarray analysis were verified by RT-PCR (*Figure 4—figure supplement 1*). The group of genes encoding serpins, as well as involucrin have been shown to play a role in keratinocyte differentiation (*Henry, 2012*; *Toulza et al., 2007*) and have been associated with psoriasis (*Roberson and Bowcock, 2010*; *Suárez-Fariñas et al., 2012*; *Wolf et al., 2012*), belonging to the 'psoriasis transcriptome' identified by *Tian et al., 2012*. In addition, dithranol downregulated expression of elevated *IL36A* and *IL36G* in human psoriatic skin and *IL36B (Il1f8)* in c-Jun/JunB psoriatic skin. To substantiate dithranol's effect on keratinocytes, we employed the mouse-tail test, a traditional model to quantify the effect of topical anti-psoriatics on keratinocyte differentiation by measuring degree of orthokeratosis versus parakeratosis (*Bosman et al., 1992*; *Sebök et al., 2000*; *Wu et al., 2015*). We found a strong increase in percentage of orthokeratosis (from 18.8 to 63.4%) reflecting dithranol's keratinocyte differentiation-inducing activity (*Figure 3—figure supplement 1*), consistent with previous work (*Bosman et al., 1992*; *Hofbauer et al., 1988*; *Sebök et al., 2000*; *Wrench and Britten, 1975*). Next, we performed RT-PCRs of a selected panel of keratinocyte differentiation markers, AMPs and inflammatory markers (based on our microarray data) of dithranol-treated murine tail skin. We found a strong upregulation of keratinization markers (*Flg, Krt16, Serpinb3a*) and several AMPs (*Lcn2, S100a8, S100a9, Defb3*)) (*Figure 3—figure supplement 2*). Interestingly, dithranol downregulated expression of the antimicrobial peptide *Camp/LL37*, as well as *Cxcl5*, a chemotactic factor for neutrophils.

In contrast to the effects of dithranol in the c-Jun/JunB model and mouse-tail test, this agent had no therapeutic capacity in the immunologically mediated imiquimod (IMQ) mouse model, which is often referred to as a psoriatic-like skin inflammation model (*van der Fits et al., 2009*). Indeed, dithranol treatment worsened psoriatic lesions in that model. Overall skin thickness was significantly enhanced, consistent with an increase in epidermal thickness and worsened inflammation (as measured by cellular infiltrate score) in mice treated with both IMQ and dithranol compared to IMQ-treated mice (*Figure 3—figure supplement 3*). Different set-ups (i.e. simultaneous treatment with dithranol and IMQ and pre-treatment with IMQ for 5 days followed by dithranol treatment) were tested, but similar effects were observed (data not shown). Taken together, these results demonstrate that dithranol primarily targets keratinocytes and only has delayed effects on other immune cells such as T cells belonging to the IL-17/IL-23 axis in psoriatic skin.

## Discussion

This study demonstrates that topical dithranol directly targets keratinocytes (in particular their differentiation regulators and AMPs), keratinocyte-neutrophil crosstalk and inhibits the IL-36 inflammatory loop in psoriasis, thus unraveling after over 100 years of use, the therapeutic mechanism of one of the most effective topical treatments of psoriasis. Dithranol significantly diminished mRNA expression of pro-psoriatic IL-1 family members (*IL36A, IL36G* and *IL36RN*) (*Supplementary file 1* and *Figure 4*). At day 6 after start of dithranol treatment, PASI had decreased by 33%, but at week 2–3 we saw a reduction of 58%, an effect that was paralleled by reduced expression of IL-36-related genes in psoriasis patients. The therapeutic importance of reduction of IL-1 family members in human skin was substantiated by results generated in the keratinocyte-based c-Jun/JunB mouse psoriasis model (*Zenz et al., 2005*). Expression of *IL36RN* (murine *Il1f5*) and *IL36B (Il1f8)* was significantly reduced in

**Table 2.** Top 45 differentially regulated genes (p-value<0.05, fold change >1.5) in dithranol-treated lesional skin after 6 days compared to baseline from 15 patients with psoriasis.

| Probe set ID | Gene symbol | Gene title | Day 6 vs. baseline (fold change) |
|---|---|---|---|
| 16693318 | FLG2 | Filaggrin family member 2 | 2.89 |
| 16693303 | HRNR | Hornerin | 2.64 |
| 16903552 | NEB | Nebulin | 2.18 |
| 16765017 | KRT2 | Keratin 2 | 2.04 |
| 16693308 | FLG | Filaggrin | 1.97 |
| 16761221 | CLEC2A | C-type lectin domain family 2, member A | 1.92 |
| 16730782 | ELMOD1 | ELMO/CED-12 domain containing 1 | 1.91 |
| 16735288 | OVCH2 | Ovochymase 2 | 1.83 |
| 16990787 | SPINK7 | Serine peptidase inhibitor, Kazal type 7 (putative) | 1.78 |
| 16852824 | SERPINB12 | Serpin peptidase inhibitor, clade B (ovalbumin), member 12 | 1.74 |
| 16693341 | LCE1C | Late cornified envelope 1C | 1.71 |
| 16693249 | THEM5 | Thioesterase superfamily member 5 | 1.70 |
| 16921644 | MIRLET7C | MicroRNA let-7c | 1.69 |
| 16670681 | ANXA9 | Annexin A9 | 1.68 |
| 16821186 | CLEC3A | C-type lectin domain family 3, member A | 1.68 |
| 16859090 | CASP14 | Caspase 14 | 1.68 |
| 16765005 | KRT73 | Keratin 73 | 1.66 |
| 16945497 | COL6A5 | Collagen, type VI, alpha 5 | 1.66 |
| 16976615 | SULT1E1 | Sulfotransferase family 1E, estrogen-preferring, member 1 | 1.65 |
| 16898858 | CD207 | CD207 molecule, langerin | 1.63 |
| 16861126 | UPK1A | Uroplakin 1A | 1.60 |
| 16704475 | FAM35DP | Family with sequence similarity 35, member A pseudogene | 1.60 |
| 17085015 | FRMPD1 | FERM and PDZ domain containing 1 | 1.59 |
| 16817034 | CHP2 | Calcineurin-like EF-hand protein 2 | 1.56 |
| 16671082 | LCE1A | Late cornified envelope 1A | 1.55 |
| 16671037 | LCE2D | Late cornified envelope 2D | 1.55 |
| 16748835 | PIK3C2G | Phosphatidylinositol-4-phosphate 3-kinase, catalytic subunit | 1.54 |
| 17039517 | LY6G6C | Lymphocyte antigen six complex, locus G6C | 1.53 |
| 16673713 | FMO2 | Flavin containing monooxygenase 2 (non-functional) | 1.52 |
| 16812344 | BCL2A1 | BCL2-related protein A1 | −1.52 |
| 16968213 | ANXA3 | Annexin A3 | −1.54 |
| 17104519 | RNY4P23 | RNA, Ro-associated Y4 pseudogene 23 | −1.55 |
| 16948835 | MIR1224 | MicroRNA 1224 | −1.55 |
| 16815310 | TNFRSF12A | Tumor necrosis factor receptor superfamily, member 12A | −1.60 |
| 17015084 | SERPINB1 | Serpin peptidase inhibitor, clade B (ovalbumin), member 1 | −1.67 |
| 17106398 | SLC6A14 | Solute carrier family 6 (amino acid transporter), member 14 | −1.69 |
| 16693375 | SPRR2F | Small proline-rich protein 2F | −1.69 |

*Table 2 continued on next page*

*Table 2 continued*

| Probe set ID | Gene symbol | Gene title | Day 6 vs. baseline (fold change) |
|---|---|---|---|
| 17065458 | DEFB4A | Defensin, beta 4A | −1.75 |
| 17074361 | DEFB4B | Defensin, beta 4B | −1.77 |
| 16698947 | RNU5A-8P | RNA, U5A small nuclear 8, pseudogene | −1.78 |
| 16976827 | CXCL5 | Chemokine (C-X-C motif) ligand 5 | −1.81 |
| 16976821 | PPBP | Pro-platelet basic protein (chemokine (C-X-C motif) ligand 7) | −1.82 |
| 17019056 | TREM1 | Triggering receptor expressed on myeloid cells 1 | −1.99 |
| 17050251 | SLC26A4 | Solute carrier family 26 (anion exchanger), member 4 | −2.19 |
| 16967771 | CXCL8 | Chemokine (C-X-C motif) ligand 8 | −3.36 |

dithranol-treated psoriatic c-Jun/JunB lesions compared to controls (*Figure 4*) and among the top differentially expressed genes in our human and mouse dataset were genes involved in keratinocyte and epidermal differentiation (*Table 4*, *Table 7*). Dithranol strongly reduced mRNA expression of the keratinocyte differentiation regulator involucrin (*IVL*) and members of the serpin family (*SERPBINB7, SERBINB13*) both in lesional skin of patients and c-Jun/JunB knockout mice (*Figure 4*). Moreover, dithranol downregulated expression of AMPs such as ß-defensins (*DEFB4A* and *DEFB4B)* produced by keratinocytes (*Liu et al., 2002*; *Schröder and Harder, 1999*) within 6 days and chemotactic factors for neutrophils (such as *CXCL5* and *CXCL8*) (*Albanesi et al., 2018*; *Table 2*) and neutrophilic infiltration (as determined by MPO staining) within 2–3 weeks of treatment in human psoriatic skin (*Figure 2*).

Surprisingly, there were no significant changes in overall T cell numbers including CD4+ and CD8 + T cells, as well as FoxP3 positivity indicative for regulatory T cells in the skin in the early phase (within 6 days) during dithranol treatment. Reflecting dithranol's primary effect on the epidermal compartment, the reduction of T cell numbers in the epidermis preceded that in the dermis. Whereas dithranol had decreased T cell counts in the epidermis at week 2–3, an effect on T cell numbers in the dermis was only evident at the follow-up visit, 4–6 weeks after the end of dithranol treatment (*Figure 2*), in agreement with previous reports favoring dithranol's effect on keratinocytes (*Holstein et al., 2017*; *Swinkels et al., 2002a*; *Vleuten et al., 1996*; *Yamamoto and Nishioka, 2003*).

*Vleuten et al., 1996* and *Swinkels et al., 2002a* applied immunohistochemistry to analyze differentiation and proliferation markers as well as T cell numbers in the skin and observed, as we did, a decrease in keratin 16, restoration of filaggrin and a decrease of Ki67 in the epidermis after 2 weeks of dithranol treatment. However, their data on the effect of dithranol on dermal T cell numbers at 4 weeks was controversial (*Swinkels et al., 2002a*; *Vleuten et al., 1996*). The group of Eberle recently tested the effects of dithranol using primary keratinocytes, a 3D psoriasis tissue model and some biopsy samples from psoriasis patients and reported on a reduction in Ki67 and keratin 16 positive cells in the epidermis using immunostaining and an inhibition of the antimicrobial peptide *DEFB4* using qPCR analysis. However, based on their observations, they concluded that dithranol's anti-psoriatic effects cannot be explained by direct effects on keratinocyte differentiation or cytokine expression (*Holstein et al., 2017*).

Our genome-wide expression analysis indicates that dithranol primarily targets keratinocytes and that this is crucial for response to treatment, considering that differentially regulated genes in histological responders compared to non-responders belonged to pathways like keratinocyte differentiation, cornification and keratin filament formation (*Supplementary file 3*). The importance of dithranol's direct effect on keratinocytes has been further substantiated by our findings generated using the mouse-tail model, a simple in vivo model to analyze effects of topical preparations on keratinocyte differentiation and parakeratosis (*Bosman et al., 1992*; *Sebök et al., 2000*; *Wu et al., 2015*). Similar to previous studies (*Bosman et al., 1992*; *Hofbauer et al., 1988*; *Sebök et al., 2000*;

**Table 3.** Top 45 differentially regulated genes in dithranol-treated lesional skin at end of treatment compared to baseline from 15 patients with psoriasis (p<0.05, fold change >1.5).

| Probe set ID | Gene symbol | Gene title | End of treatment vs. baseline (fold change) |
|---|---|---|---|
| 16947045 | AADAC | Arylacetamide deacetylase | 3.07 |
| 16765005 | KRT73 | Keratin 73 | 2.31 |
| 16976868 | BTC | Betacellulin | 2.27 |
| 16924792 | CLDN8 | Claudin 8 | 2.11 |
| 17125092 | CNTNAP3 | Contactin associated protein-like 3 | 2.09 |
| 17097358 | SLC46A2 | Solute carrier family 46, member 2 | 2.07 |
| 16780133 | SLITRK6 | SLIT and NTRK-like family, member 6 | 2.00 |
| 16834436 | RAMP2 | Receptor (G protein-coupled) activity modifying protein 2 | 1.96 |
| 17123970 | ZDHHC11B | Zinc finger, DHHC-type containing 11B | 1.88 |
| 17104313 | AR | Androgen receptor | 1.85 |
| 16951140 | MIR548I2 | MicroRNA 548i-2 | 1.85 |
| 17063366 | ATP6V0A4 | ATPase, H+ transporting, lysosomal V0 subunit a4 | 1.84 |
| 16974224 | MIR548I2 | MicroRNA 548i-2 | 1.83 |
| 16687914 | CYP2J2 | Cytochrome P450, family 2, subfamily J, polypeptide 2 | 1.82 |
| 17125106 | CNTNAP3B | Contactin associated protein-like 3B | 1.80 |
| 16903552 | NEB | Nebulin | 1.80 |
| 17125094 | CNTNAP3B | Contactin associated protein-like 3B | 1.80 |
| 16770284 | TMEM116 | Transmembrane protein 116 | 1.79 |
| 16765041 | KRT77 | Keratin 77 | 1.79 |
| 17123972 | ZDHHC11B | Zinc finger, DHHC-type containing 11B | 1.78 |
| 16688210 | MIR3671 | MicroRNA 3671 | 1.78 |
| 17125218 | CNTNAP3 | Contactin associated protein-like 3 | 1.74 |
| 16764923 | KRT6A | Keratin 6A | −3.69 |
| 16767261 | IL22 | Interleukin 22 | −3.87 |
| 17065453 | DEFB103A | Defensin, beta 103A | −3.94 |
| 17074366 | DEFB103A | Defensin, beta 103A | −3.94 |
| 16671027 | LCE3C | Late cornified envelope 3C | −4.14 |
| 16743751 | MMP12 | Matrix metallopeptidase 12 (macrophage elastase) | −4.42 |
| 16979444 | TNIP3 | TNFAIP3 interacting protein 3 | −4.48 |
| 17106398 | SLC6A14 | Solute carrier family 6 (amino acid transporter), member 14 | −4.60 |
| 16686734 | CYP4Z2P | Cytochrome P450, family 4, subfamily Z, polypeptide 2 | −4.89 |
| 16671144 | S100A7A | S100 calcium binding protein A7A | −5.00 |
| 16764907 | KRT6C | Keratin 6C | −5.01 |
| 16730157 | HEPHL1 | Hephaestin-like 1 | −5.15 |
| 16967831 | EPGN | Epithelial mitogen | −5.31 |
| 16842517 | NOS2 | Nitric oxide synthase 2, inducible | −5.38 |
| 16693409 | S100A12 | S100 calcium binding protein A12 | −6.12 |
| 16813112 | RHCG | Rh family, C glycoprotein | −6.18 |
| 16738803 | TCN1 | Transcobalamin I (vitamin B12 binding protein, R binder family) | −6.45 |

*Table 3 continued on next page*

*Table 3 continued*

| Probe set ID | Gene symbol | Gene title | End of treatment vs. baseline (fold change) |
|---|---|---|---|
| 16924785 | CLDN17 | Claudin 17 | −8.09 |
| 16693365 | SPRR2C | Small proline-rich protein 2C (pseudogene) | −8.72 |
| 16967771 | CXCL8 | Chemokine (C-X-C motif) ligand 8 | −9.14 |
| 17074361 | DEFB4B | Defensin, beta 4B | −9.39 |
| 16693375 | SPRR2F | Small proline-rich protein 2F | −10.12 |
| 16884602 | IL36A | Interleukin 36, alpha | −10.50 |

*Wrench and Britten, 1975*), we observed a strong increase in orthokeratosis after dithranol application in the mouse-tail test, reflecting its keratinocyte differentiation-inducing activity (*Figure 3—figure supplement 1*). Our transcriptional analysis indicated that the effect of dithranol in the mouse-tail test was linked to a strong upregulation of keratinocyte differentiation markers and several AMPs, while the pro-psoriatic antimicrobial peptide *Camp/LL37* was downregulated, as well as *Cxcl5*, a chemotactic factor for neutrophils (*Figure 3—figure supplement 2*), but not other pro-psoriatic AMPs (*Fritz et al., 2017*; *Wang et al., 2018*) such as *Defb3, S100a8* or *S100a9*. Although the

**Table 4.** Top 20 significantly enriched pathways as determined by Gene Ontology (GO) enrichment analysis in dithranol-treated lesional skin after 6 days compared to baseline from 15 patients with psoriasis.
(GO was done using Cytoscape software [*Bindea et al., 2009*; *Shannon et al., 2003*]; a.o. = among others).

| GOID | GO term | P-Value | % Associated Genes | Associated genes found |
|---|---|---|---|---|
| GO:0001533 | Cornified envelope | 3.8E-12 | 10.45 | FLG, HRNR, KRT2, LCE1A, LCE1C, LCE2D, SPRR2F |
| GO:0031424 | Keratinization | 7.7E-12 | 3.93 | CASP14, FLG, HRNR, KRT2, KRT73, LCE1A, LCE1C, LCE2D, SPRR2F |
| GO:0030216 | Keratinocyte differentiation | 79.0E-12 | 3.02 | CASP14, FLG, HRNR, KRT2, KRT73, LCE1A, LCE1C, LCE2D, SPRR2F |
| GO:0018149 | Peptide cross-linking | 300.0E-12 | 9.84 | FLG, KRT2, LCE1A, LCE1C, LCE2D, SPRR2F |
| GO:0070268 | Cornification | 12.0E-9 | 5.31 | CASP14, FLG, KRT2, KRT73, LCE1A, SPRR2F |
| GO:0004168 | Receptor CXCR2 binds ligands CXCL1 to 7 | 1.0E-6 | 33.33 | CXCL5, CXCL8, PPBP |
| GO:0042379 | Chemokine receptor binding | 5.4E-6 | 6.25 | CXCL5, CXCL8, DEFB4A, PPBP |
| GO:0045236 | CXCR chemokine receptor binding | 6.0E-6 | 18.75 | CXCL5, CXCL8, PPBP |
| GO:0061436 | Establishment of skin barrier | 18.0E-6 | 13.04 | FLG, FLG2, HRNR |
| GO:0033561 | Regulation of water loss via skin | 21.0E-6 | 12.00 | FLG, FLG2, HRNR |
| GO:0004657 | IL-17 signaling pathway | 21.0E-6 | 4.30 | CXCL5, CXCL8, DEFB4A, DEFB4B |
| GO:0007874 | Keratin filament formation | 21.0E-6 | 4.17 | FLG, KRT2, KRT73, SPRR2F |
| GO:0030593 | Neutrophil chemotaxis | 21.0E-6 | 4.17 | CXCL5, CXCL8, PPBP, TREM1 |
| GO:1990266 | Neutrophil migration | 27.0E-6 | 3.85 | CXCL5, CXCL8, PPBP, TREM1 |
| GO:0071621 | Granulocyte chemotaxis | 38.0E-6 | 3.31 | CXCL5, CXCL8, PPBP, TREM1 |
| GO:1902622 | Regulation of neutrophil migration | 45.0E-6 | 7.50 | CXCL5, CXCL8, PPBP |
| GO:0071622 | Regulation of granulocyte chemotaxis | 71.0E-6 | 5.66 | CXCL5, CXCL8, PPBP |
| GO:0030104 | Water homeostasis | 130.0E-6 | 4.00 | FLG, FLG2, HRNR |
| GO:0002690 | Positive regulation of leukocyte chemotaxis | 120.0E-6 | 3.30 | CXCL5, CXCL8, PPBP |
| GO:0004867 | Serine-type endopeptidase inhibitor activity | 79.0E-6 | 3.00 | SERPINB1, SERPINB12, SPINK7 |

**Table 5.** Top 20 significantly enriched pathways as determined by Gene Ontology (GO) enrichment analysis in dithranol-treated lesional skin at end of treatment compared to baseline from 15 patients with psoriasis.
(GO was done using Cytoscape software; *Bindea et al., 2009*; *Shannon et al., 2003* a.o. = among others).

| GOID | GO term | P-Value | % Associated Genes | Associated genes found |
|---|---|---|---|---|
| GO:0006954 | Inflammatory response | 17.0E-18 | 6.93 | IL17A, IL1B, IL20, IL22, IL36A, IL36G, IL36RN, a.o. |
| GO:0006952 | Defense response | 52.0E-18 | 4.60 | CXCL8, CXCL9, DEFB103A, DEFB4A, IFNG, IL17A, IL1B, IL20, IL22, IL36A, IL36G, IL36RN, IL4R, IRAK2, IRF1, KRT16, a.o. |
| GO:0051707 | Response to other organism | 56.0E-18 | 6.02 | CCL2, CCL20, CCL22, CD24, CD80, COTL1, CXCL13, CXCL16, CXCL8, CXCL9, DDX21, DEFB103A, DEFB4A, a.o. |
| GO:0050663 | Cytokine secretion | 480.0E-18 | 13.78 | IFNG, IL17A, IL1B, IL26, IL36RN, IL4R, LYN, MMP12, NOS2, PAEP, PANX1, PNP, S100A12, S100A8, S100A9, a.o. |
| GO:0001816 | Cytokine production | 680.0E-18 | 6.76 | IDO1, IFNG, IL17A, IL1B, IL26, IL36A, IL36RN, IL4R, IRF1, LTF, LYN, MB21D1, MMP12, NOS2, a.o. |
| GO:0012501 | Programmed cell death | 1.0E-15 | 4.05 | CASP5, CASP7, CCL2, CCL21, CD24, CD274, CD38, IFNG, IL17A, IL1B, IRF1, IVL, KLK13, KRT16, KRT17, KRT31, KRT6A, KRT6C, KRT73, KRT74, KRT77, a.o. |
| GO:0051240 | Positive regulation of multicellular organismal process | 1.5E-15 | 4.57 | CXCL17, CXCL8, FBN2, FERMT1, GBP5, GPR68, HPSE, HRH2, IDO1, IFNG, IL17A, IL1B, IL20, IL26, IL36A, S100A8, S100A9, SERPINB3, SERPINB7, a.o. |
| GO:0001817 | Regulation of cytokine production | 2.0E-15 | 7.04 | CCL2, CCL20, CD24, CD274, CD80, CD83, CXCL17, IFNG, IL17A, IL1B, IL26, IL36A, IL36RN, IL4R, IRF1, TNFRSF9, WNT5A, a.o. |
| GO:0002237 | Response to molecule of bacterial origin | 15.0E-15 | 9.22 | S100A8, S100A9, SELE, SOD2, TIMP4, TNFRSF9, TNIP3, WNT5A, ZC3H12A, a.o. |
| GO:0070268 | Cornification | 100.0E-15 | 17.70 | DSC2, DSG3, IVL, KLK13, KRT16, KRT17, KRT31, KRT6A, KRT6C, KRT73, KRT74, KRT77, PI3, SPRR2A, SPRR2B, SPRR2D, a.o. |
| GO:0043588 | Skin development | 190.0E-15 | 8.17 | IL20, IVL, KLK13, KRT16, KRT17, KRT31, KRT6A, KRT6C, KRT73, KRT74, KRT77, LCE3A, LCE3C, LCE3E, a.o. |
| GO:0008544 | Epidermis development | 220.0E-15 | 7.63 | DSC2, DSG3, EPHA2, FERMT1, FOXE1, FURIN, HPSE, IL20, IVL, KLK13, KRT16, KRT17, KRT31, KRT6A, KRT6C, KRT73, KRT74, KRT77, a.o. |
| GO:0009617 | Response to bacterium | 240.0E-15 | 6.63 | CCL2, CCL20, CD24, CD80, CXCL13, CXCL16, CXCL8, CXCL9, DEFB103A, DEFB4A, S100A12, S100A8, S100A9, TREM1, a.o. |
| GO:0001819 | Positive regulation of cytokine production | 1.4E-12 | 7.89 | CCL2, CCL20, CD274, CD80, CD83, CXCL17, FERMT1, GBP5, HPSE, IDO1, IFNG, IL17A, IL1B, IL26, IL36A, a.o. |
| GO:0050707 | Regulation of cytokine secretion | 2.4E-12 | 13.10 | AIM2, CASP5, CD274, FERMT1, GBP1, IFNG, IL17A, IL1B, IL26, IL36RN, IL4R, LYN, MMP12, PAEP, PANX1, a.o. |
| GO:0005125 | Cytokine activity | 4.4E-12 | 10.64 | CCL20, CCL21, CCL22, CCL4L2, CXCL13, CXCL16, CXCL8, CXCL9, FAM3D, IFNG, IL17A, IL1B, IL20, IL22, IL26, a.o. |
| GO:0031424 | Keratinization | 21.0E-12 | 10.48 | IVL, KLK13, KRT16, KRT17, KRT31, KRT6A, KRT6C, KRT73, KRT74, KRT77, LCE3A, LCE3C, LCE3E, PI3, SPRR2A, SPRR2B, a.o. |
| GO:0002790 | Peptide secretion | 97.0E-12 | 6.25 | CD274, CD38, DOC2B, FAM3D, FERMT1, GBP1, GBP5, GLUL, GPR68, IFNG, IL17A, IL1B, IL26, IL36RN, IL4R, LYN, MMP12, NOS2, TREM1, WNT5A, a.o. |

*Table 5 continued on next page*

*Table 5 continued*

| GOID | GO term | P-Value | % Associated Genes | Associated genes found |
|------|---------|---------|--------------------|-----------------------|
| GO:0032940 | Secretion by cell | 140.0E-12 | 4.04 | AIM2, AMPD3, CASP5, IL36RN, IL4R, LCN2, LRG1, LTF, LYN, MMP12, NOS2, NR1D1, NR4A3, OLR1, PAEP, PANX1, PLA2G3, a.o. |
| GO:0009913 | Epidermal cell differentiation | 190.0E-12 | 7.91 | DSC2, DSG3, EPHA2, FURIN, IL20, IVL, KLK13, KRT16, KRT17, KRT31, KRT6A, KRT6C, KRT73, LCE3E, PI3, SLITRK6, SPRR2A, SPRR2B, SPRR2D, SPRR2E, a.o. |

mouse-tail test has evidently limitations since the disturbed cell differentiation of this model only reflects one of many aspects of psoriasis, it supports the primary effect of dithranol on keratinocytes with induced induction of orthokeratosis.

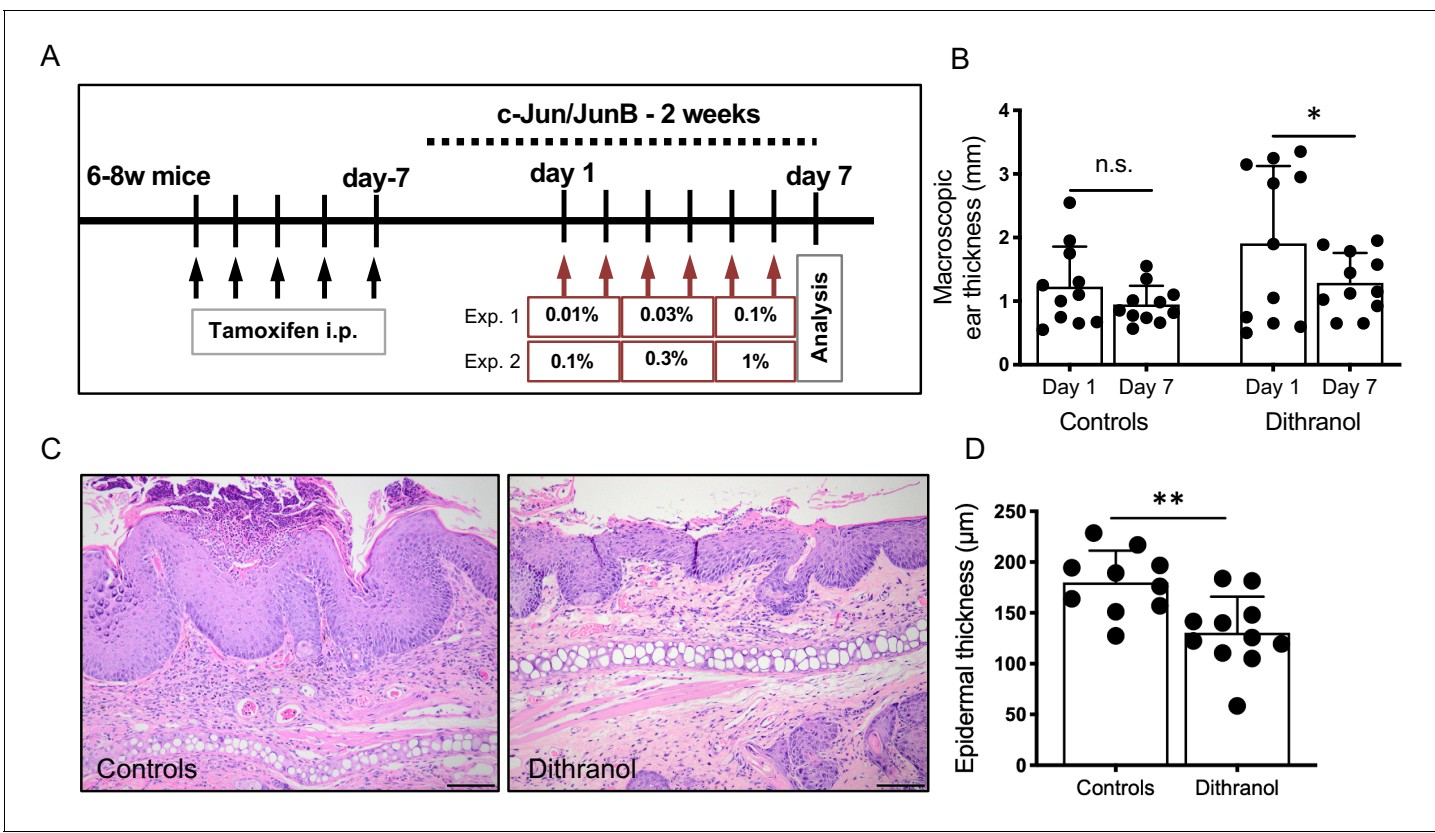

**Figure 3.** Topical application of dithranol ameliorates psoriasis-like skin lesions in c-Jun/JunB knockout mice. (**A**) Schematic representation of experimental set-up. Red arrows indicate dithranol application in series of increasing concentrations. (Exp. 1 and 2 = experiment 1 and 2) (**B**) Macroscopic ear thickness on day 1 compared to day 7. Dithranol treatment led to a significant reduction in ear thickness. Controls (n = 11), dithranol group (n = 11); Paired t-test was used for statistics. (**C**) Representative H and E images of untreated and dithranol-treated ears. (**D**) Dithranol treatment led to a significant reduction in epidermis thickness. Controls (n = 10), dithranol group (n = 11); unpaired t-test was used for statistics. Data from the two experiments was pooled (**D**). Bars represent mean ± SD; n.s. = not significant; *$p \leq 0.05$; **$p \leq 0.01$; ***$p \leq 0.001$; ****$p \leq 0.0001$; scale bar = 100 μm. The online version of this article includes the following source data and figure supplement(s) for figure 3:

**Source data 1.** Values displayed in scatter plots shown in *Figure 3*.
**Figure supplement 1.** Strong inducing effect of dithranol in the mouse-tail test.
**Figure supplement 2.** Gene expression analysis by RT-PCR of keratinization markers (Flg, Ivl, Krt16, Lce3e, Serpinb3a), AMPs (Lcn2, S100a8, S100a9, Camp, Defb1, Defb3) and inflammatory markers (Il1b, Il17, Il22, Cxcl1, Cxcl5) of dithranol- and vehicle-treated murine tail skin.
**Figure supplement 3.** Dithranol causes exacerbation of psoriasis-like skin lesions in imiquimod mouse model.

**Table 6.** Top 45 differentially regulated genes in dithranol-treated psoriatic skin of c-Jun/JunB knockout mice compared to that of vehicle-treated controls (p<0.05, fold change (FC) >1.5).

| Probe set ID | Gene symbol | Gene title | Fold change |
| --- | --- | --- | --- |
| TC0Y00000006.mm.2 | Eif2s3y | Eukaryotic translation initiation factor 2, subunit 3, structural gene Y-linked | 10,37 |
| TC0Y00000233.mm.2 | Uty | Ubiquitously transcribed tetratricopeptide repeat gene, Y chromosome | 6,38 |
| TC0Y00000235.mm.2 | Ddx3y | DEAD (Asp-Glu-Ala-Asp) box polypeptide 3, Y-linked | 5,40 |
| TC0900000047.mm.2 | Mmp3 | Matrix metallopeptidase 3 | 3,64 |
| TC0Y00000079.mm.2 | Ssty2 | Spermiogenesis specific transcript on the Y 2 | 3,31 |
| TC0100001632.mm.2 | Ifi209 | Interferon activated gene 209 | 3,23 |
| TC0500002755.mm.2 | Cxcl9 | Chemokine (C-X-C motif) ligand 9 | 2,85 |
| TC0300003133.mm.2 | Ifi44 | Interferon-induced protein 44 | 2,69 |
| TC0100003591.mm.2 | Ifi213 | Interferon activated gene 213 | 2,53 |
| TC0100001634.mm.2 | Ifi208 | Interferon activated gene 208 | 2,49 |
| TC1900000217.mm.2 | Ms4a4c | Membrane-spanning 4-domains, subfamily A, member 4C | 2,45 |
| TC0300002684.mm.2 | Chil3 | Chitinase-like 3 | 2,39 |
| TC0100003550.mm.2 | Slamf7 | SLAM family member 7 | 2,33 |
| TC0500000922.mm.2 | Cxcl13 | Chemokine (C-X-C motif) ligand 13 | 2,26 |
| TC1900000500.mm.2 | Ifit2 | Interferon-induced protein with tetratricopeptide repeats 2 | 2,20 |
| TC0500002840.mm.2 | Plac8 | Placenta-specific 8 | 2,20 |
| TC0700001630.mm.2 | Adm | Adrenomedullin | 2,15 |
| TC1900000501.mm.2 | Ifit3 | Interferon-induced protein with tetratricopeptide repeats 3 | 2,13 |
| TC1800000610.mm.2 | Iigp1 | Interferon inducible GTPase 1 | 2,07 |
| TC0300001446.mm.2 | Gbp3 | Guanylate binding protein 3 | 2,07 |
| TC0300003134.mm.2 | Ifi44l | Interferon-induced protein 44 like | 2,05 |
| TC1900000502.mm.2 | Ifit3b | Interferon-induced protein with tetratricopeptide repeats 3B | 2,02 |
| TC0400003296.mm.2 | Skint5 | Selection and upkeep of intraepithelial T cells 5 | −2,37 |
| TC0300000482.mm.2 | Aadac | Arylacetamide deacetylase | −2,37 |
| TC1100000469.mm.2 | Fndc9 | Fibronectin type III domain containing 9 | −2,39 |
| TC0300002399.mm.2 | Lce6a | Late cornified envelope 6A | −2,42 |
| TC0300002409.mm.2 | Lce1i | Late cornified envelope 1I | −2,44 |
| TC0300002412.mm.2 | Kprp | Keratinocyte expressed, proline-rich | −2,44 |
| TC0300000846.mm.2 | Hrnr | Hornerin | −2,48 |
| TC0300002407.mm.2 | Lce1g | Late cornified envelope 1G | −2,49 |
| TC1600000324.mm.2 | Fetub | Fetuin beta | −2,53 |
| TC1300001377.mm.2 | Akr1c18 | Aldo-keto reductase family 1, member C18 | −2,56 |
| TC0300002406.mm.2 | Lce1f | Late cornified envelope 1F | −2,57 |
| TC1500001714.mm.2 | Slurp1 | Secreted Ly6/Plaur domain containing 1 | −2,57 |
| TC1500002344.mm.2 | Gsdmc2 | Gasdermin C2 | −2,59 |
| TC1000000145.mm.2 | Il20ra | Interleukin 20 receptor, alpha | −2,66 |
| TC0700000443.mm.2 | Cyp2b19 | Cytochrome P450, family 2, subfamily b, polypeptide 19 | −2,70 |
| TC0100000103.mm.2 | Ly96 | Lymphocyte antigen 96 | −2,89 |

*Table 6 continued on next page*

*Table 6 continued*

| Probe set ID | Gene symbol | Gene title | Fold change |
|---|---|---|---|
| TC0700000778.mm.2 | Klk5 | Kallikrein related-peptidase 5 | −3,15 |
| TC0300003252.mm.2 | Clca3a2 | Chloride channel accessory 3A2 | −3,20 |
| TC0900002312.mm.2 | Elmod1 | ELMO/CED-12 domain containing 1 | −3,20 |
| TC0300002401.mm.2 | Lce1a1 | Late cornified envelope 1A1 | −3,26 |
| TC0700000484.mm.2 | Fcgbp | Fc fragment of IgG binding protein | −3,83 |
| TC0300002405.mm.2 | Lce1e | Late cornified envelope 1E | −4,16 |
| TC1500002274.mm.2 | Krt2 | Keratin 2 | −5,95 |

In contrast to its therapeutic effect in both keratinocyte-driven psoriasis models, the c-Jun/JunB model and mouse-tail test, dithranol did aggravate psoriatic lesions in the imiquimod model that has been solely shown to be immunologically mediated and dependent on the IL-17/IL-23 axis (*van der Fits et al., 2009*). In the latter model, biologics such as etanercept and anti-IL-17a agents (*Liu et al., 2018*), topical steroids (*Sun et al., 2014*) and vitamin d3 analogues (*Germán et al., 2019*) but also UVB and PUVA (*Shirsath et al., 2018*) were shown to have a beneficial effect. In contrast, dithranol significantly enhanced overall macroscopic skin thickness, consistent with a slight increase in epidermal hyperplasia and worsened inflammation (as measured by the density of cellular infiltrate in the dermis) (*Figure 3—figure supplement 3*). Dithranol treatment of healthy murine skin led to similar effects upon irritation, as it increased epidermal thickness and cellular infiltrate of the skin (data not shown). However, the irritant effect of dithranol may remain without functional anti-psoriatic relevance in human psoriasis (as indicated by the clinical results depicted in *Figure 1—figure supplement 2* and discussed below) but might be crucial in the agent's therapeutic action in alopecia areata (*Nasimi et al., 2019*; *Ngwanya et al., 2017*).

Together, these data unambiguously demonstrate that dithranol directly acts on keratinocytes, their crosstalk with neutrophils and IL-36 signaling, with AMPs being the potential link (*Figure 5*). This goes also in line with the observation that Langerhans cells as another type of immune cells showed a delayed response to dithranol, with no changes during treatment and an evident increase in numbers only later on (at the follow-up visit 4–6 weeks after end of treatment). These results are consistent with previous studies performed by Swinkels et al., showing that there was no significant change in T cells or Langerhans cells during twelve days of dithranol treatment (*Swinkels et al., 2002a*).

Our findings on dithranol's effect on members of the IL-1 family, being beneficial for its anti-psoriatic efficacy, are well in line with recent work on blockade of IL-36 pathway as a novel strategy for the treatment of pustular psoriasis (*Bachelez et al., 2019*) as well as plaque-type psoriasis (*Benezeder and Wolf, 2019*; *Mahil et al., 2017*). Notably, human keratinocytes express IL-1 family members (IL-36α, IL-36β, IL-36γ, IL-36Ra) and their receptor IL-36R (*D'Erme et al., 2015*; *Foster et al., 2014*; *Johnston et al., 2017*). Furthermore, normal human keratinocytes show increased expression of IL-1 group mRNA after treatment with psoriasis-associated cytokines (TNFα, IL-1α, IL-17,IL-22) (*Johnston et al., 2011*). Genome-wide association studies revealed that deficiency in interleukin-36 receptor antagonist due to IL36RN mutations was associated with generalized pustular psoriasis (GPP) (*Marrakchi et al., 2011*; *Sugiura et al., 2013*). Supporting this notion, blocking IL-36 receptor was effective in reducing epidermal hyperplasia and showed comparable effects to etanercept in a psoriatic skin xenotransplantation model (*Blumberg et al., 2010*). Furthermore, successful treatment of psoriasis with the anti-psoriatic standard treatment etanercept is accompanied by a decrease in *IL36A, IL36G* and *IL36RN* expression (*Johnston et al., 2011*). A recent clinical phase one study provided proof-of-concept for the efficacy of BI 655130, a monoclonal antibody against the interleukin-36 receptor in the treatment of generalized pustular psoriasis (*Bachelez et al., 2019*). Intriguingly, topical short-contact dithranol therapy is efficacious not only in plaque-type psoriasis, but under certain conditions (i.e. after stabilization of disease activity with bland emollients) reportedly also in pustular psoriasis (*Farber and Nall, 1993*; *Gerritsen, 1999*). In this neutrophilic-driven condition, in which one might expect that dithranol worsens a heavy inflammatory state, it may have

**Table 7.** Top 20 significantly enriched pathways as determined by Gene Ontology (GO) enrichment analysis in dithranol-treated psoriatic skin of c-Jun/JunB knockout mice compared to that of vehicle-treated controls.
(GO was done using Cytoscape software *Bindea et al., 2009*; *Shannon et al., 2003*; a.o. = among others).

| Go id | GO term | P-Value | % Associated Genes | Associated genes found |
|---|---|---|---|---|
| GO:0043588 | Skin development | 14,0E-18 | 11,11 | Casp14, Cldn1, Flg2, Gjb3, Hrnr, Ivl, Krt2, Lce1a1, Lce1a2, Lce1l, Lce1m, Lor, Pou2f3, Ptgs1, Scel, Tfap2b, a.o. |
| GO:0008544 | Epidermis development | 73,0E-15 | 9,18 | Acer1, Alox8, Casp14, Cnfn, Cst6, Dnase1l2, Hrnr, Ivl, Krt2, Lce1c, Lce1d, Lce1e, Lce1f, Lce1g, Lce1h, Lce1i, Lce1j, a.o. |
| GO:0071345 | Cellular response to cytokine stimulus | 1,7E-12 | 6,26 | Ccdc3, Ccl2, Ccl5, Ccr9, Cxcl13, Cxcl9, Edn1, Il18, Il1f5, Il1f8, Il1rl1, Il20ra, Stat2, a.o. |
| GO:0034097 | Response to cytokine | 7,1E-12 | 5,62 | Cd38, Chad, Cldn1, Csf3, Edn1, Gbp2, Gbp3, Gbp4, Gbp7, Gbp8, Gbp9, Gm4951, Ifi203, Ifi204, Ifi209, Ifit1, Ifit2, Ifit3, Ifit3b, Xaf1, a.o. |
| GO:0035456 | Response to interferon-beta | 12,0E-12 | 25,00 | Gbp2, Gbp3, Gbp4, Gm4951, Ifi203, Ifi204, Ifi209, Ifit1, Ifit3, Iigp1, Irgm1, Xaf1 |
| GO:0006952 | Defense response | 63,0E-12 | 4,11 | Ccl2, Ccl5, Cd180, Cd59a, Cxcl13, Cxcl9, Cybb, Defb6, Drd1, Herc6, Hp, Il18, Il1f5, Il1f8, Il1rl1, Irgm1, Kalrn, Klk5, a.o. |
| GO:0030855 | Epithelial cell differentiation | 430,0E-12 | 5,51 | Casp14, Cdkn1a, Cldn1, Cnfn, Dlx3, Dnase1l2, Gsdmc2, Gstk1, Hrnr, Ivl, Klf15, Krt2, Lce1a1, Lor, Pou2f3, a.o. |
| GO:0020005 | Symbiont-containing vacuole membrane | 1,4E-9 | 66,67 | Gbp2, Gbp3, Gbp4, Gbp7, Gbp9, Iigp1 |
| GO:0044216 | Other organism cell | 5,6E-9 | 30,77 | C4b, Gbp2, Gbp3, Gbp4, Gbp7, Gbp9, Iigp1, Tap1 |
| GO:0044406 | Adhesion of symbiont to host | 120,0E-9 | 37,50 | Ace2, Gbp2, Gbp3, Gbp4, Gbp7, Gbp9 |
| GO:0045087 | Innate immune response | 390,0E-9 | 4,35 | Ccl2, Ccl5, Cd180, Cfb, Cldn1, Ifit3, Iigp1, Il1f5, Il1f8, Irgm1, Lbp, Ly96, Mx1, Parp9, Sla, Slamf6, Slamf7, Stat2, Tlr7, Trim62, a.o. |
| GO:0034341 | Response to interferon-gamma | 1,0E-6 | 10,58 | Ccl2, Ccl5, Cldn1, Edn1, Gbp2, Gbp3, Gbp4, Gbp7, Gbp8, Gbp9, Parp9 |
| GO:0006954 | Inflammatory response | 5,9E-6 | 4,29 | C3, C4b, Ccl2, Ccl5, Cd180, Cd59a, Chil3, Crip2, Ctla2a, Cxcl13, Cxcl9, Cybb, Hp, Il18, Il1f5, Il1f8, Il1rl1, Lbp, Ly96, a.o. |
| GO:0042832 | Defense response to protozoan | 9,4E-6 | 19,35 | Gbp2, Gbp3, Gbp4, Gbp7, Gbp9, Iigp1 |
| GO:0035457 | Cellular response to interferon-alpha | 21,0E-6 | 36,36 | Ifit1, Ifit2, Ifit3, Ifit3b |
| GO:0030414 | Peptidase inhibitor activity | 22,0E-6 | 6,19 | C3, C4b, Cst6, Ctla2b, Fetub, R3hdml, Serpinb12, Serpinb13, Serpinb2, Serpinb7, Spink14, Tfap2b, Wfdc12, Wfdc5 |
| GO:0098542 | Defense response to other organism | 34,0E-6 | 4,05 | Adm, Ccl5, Cxcl13, Cxcl9, Defb6, Gbp2, Gbp3, Hp, Ifit1, Ifit2, Ifit3, Ifit3b, Iigp1, Il1f5, Klk5, Lbp, Mx1, Oas1f, Plac8, Stat2, Tlr7, Wfdc12, a.o. |
| GO:0031424 | Keratinization | 43,0E-6 | 15,00 | Casp14, Cnfn, Hrnr, Ivl, Krt2, Lor |
| GO:0044403 | Symbiosis, encompassing mutualism through parasitism | 55,0E-6 | 4,11 | Ace2, Acta2, Atg16l2, Ccl5, Cxcl9, Gbp2, Gbp3, Gbp4, Gbp7, Gbp9, Ifit1, Ifit2, Ifit3, Ifit3b, Lbp, Mx1, Oas1f, Pou2f3, Rab9, Stat2, Tap1, Tlr7, Trim62 |

beneficial capacity by targeting the IL-36 pathway and neutrophils, both playing a most crucial role in this type of psoriasis (*Bachelez, 2018*; *Bachelez et al., 2019*; *Johnston et al., 2017*; *Marrakchi et al., 2011*; *Sugiura et al., 2013*).

As outlined above, we observed a strong effect of dithranol on keratinocyte-neutrophil crosstalk. Importantly, early response to anti-IL-17a blockade, as one of the most effective biological treatments currently available, has been linked to inhibition of keratinocyte-neutrophil crosstalk (*Reich et al., 2015*). Similar to our observation, anti-IL-17a treatment with secukinumab significantly reduced epidermal hyperproliferation after 2 weeks, decreased mRNA expression of keratinocyte-derived chemotactic factors and led to a strong decrease in IL-17a positive neutrophils. Apparently,

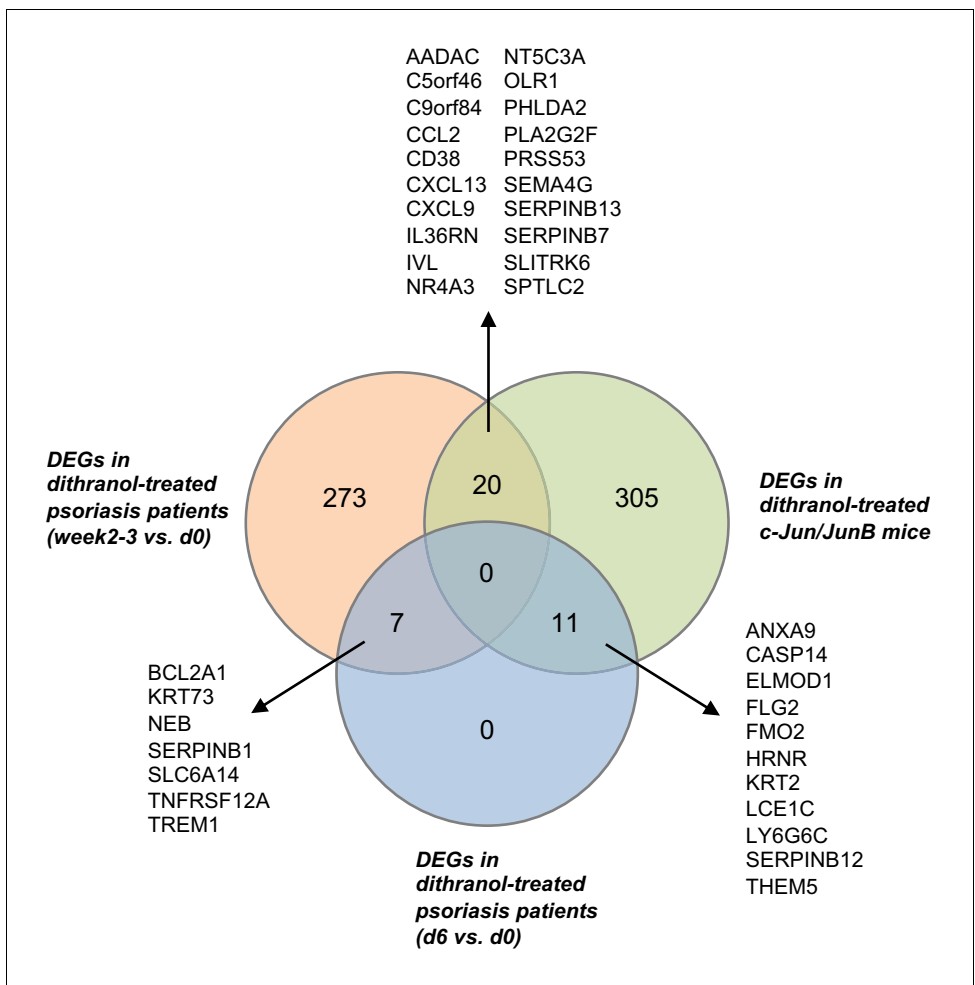

**Figure 4.** Venn diagram (created using InteractiVenn *Heberle et al., 2015*) showing comparison and overlap between differentially expressed genes (DEGs) in dithranol-treated human psoriatic skin at week 2–3 vs. day 0, DEGs in dithranol-treated human psoriatic skin at day 6 vs. day 0 and DEGs in dithranol-treated c-Jun/JunB psoriatic skin.

The online version of this article includes the following figure supplement(s) for figure 4:

**Figure supplement 1.** Verification of microarray gene expression analysis: Gene expression analysis by RT-PCR of selected genes based on microarray data in dithranol-treated c-Jun/JunB knockout mice vs. vehicle-treated controls.

disrupting the crosstalk between keratinocytes and neutrophils may be a crucial early effect in the clinical efficacy of secukinumab in psoriasis (*Reich et al., 2015*). In addition, decreased expression of AMPs like ß-defensin and S100 proteins was observed after only 1 week of anti-IL-17a treatment with secukinumab (*Krueger et al., 2019*) and after 2 weeks of ixekizumab treatment (*Krueger et al., 2012*), well in line with our observed early effect of dithranol on AMPs. Krueger et al. concluded that clinical efficacy of anti-IL-17a treatment is closely linked to early inhibition of keratinocyte-derived products such as chemokines and AMPs. Together this suggests that dithranol's direct action on keratinocytes at the molecular level may disrupt IL-17 pathway dysregulation (without directly blocking IL-17 or its receptor), leading in turn to similar downstream effects as treatment with anti-IL-17 antagonists.

Our study also negates the paradigm that dithranol-induced irritation is crucial for its anti-psoriatic action (*Gerritsen, 1999*; *Kucharekova et al., 2005*; *Prins et al., 1998*; *van de Kerkhof, 1991*; *Wiegrebe and Müller, 1995*). As depicted in *Figure 1—figure supplement 2*, there was no correlation between dithranol-induced perilesional as well as lesional erythema and its anti-psoriatic effect,

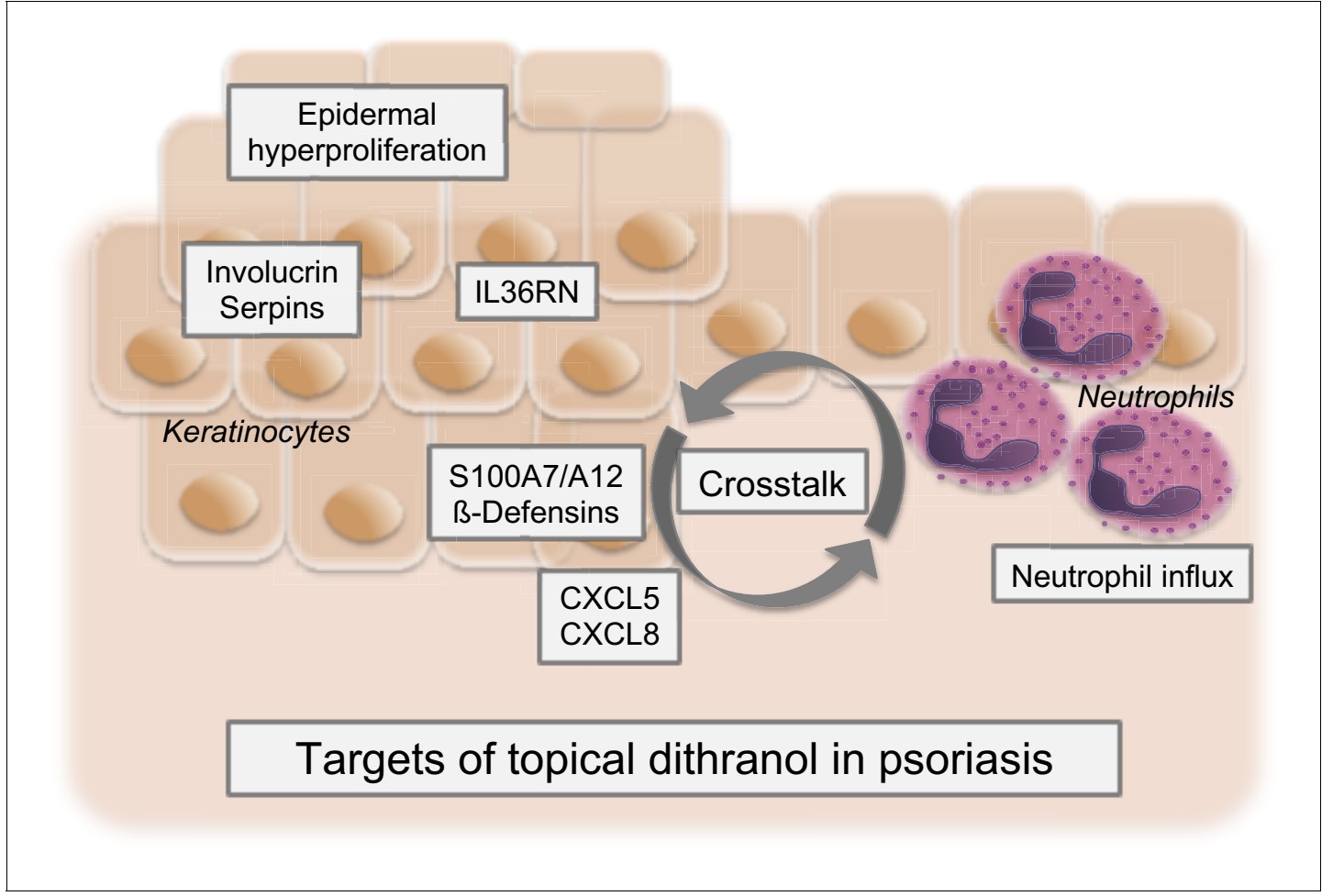

**Figure 5.** Proposed model of dithranol's mechanism of action in psoriasis. Dithranol decreases expression of keratinocyte differentiation regulators (involucrin and serpins), IL-36-related genes, keratinocyte-derived antimicrobial peptides (AMPs) (S100A7/A12 and ß-defensins) and chemotactic factors for neutrophils (CXCL5, CXCL8). This disrupts the crosstalk between keratinocytes and neutrophils and leads to diminished neutrophilic infiltration, thereby halting the inflammatory feedback loop in psoriasis. Together, this results in clearance of psoriatic lesions.

indicating that dithranol's irritation (*Prins et al., 1998*; *Swinkels et al., 2002c*; *Swinkels et al., 2002b*) is a treatment side-effect unrelated to its therapeutic mechanism. However, this irritant effect of dithranol may be crucial for its action in alopecia areata, a condition, in which it was shown to be greatly effective, leading to hair regrowth in a high percentage of cases (*Ngwanya et al., 2017*). Similar to other topical treatment options (*Nasimi et al., 2019*; *Yoshimasu and Furukawa, 2016*), an initial irritant reaction to dithranol is followed by regrowth of hair within weeks after treatment.

What our work does not answer, is how dithranol exactly acts at the molecular level. Cell culture studies have shown that dithranol targets mitochondria (*McGill et al., 2005*), alters cellular metabolism (*Hollywood et al., 2015*) and induces apoptosis in keratinocytes (*George et al., 2013*; *McGill et al., 2005*). Dithranol also inhibits human monocytes in vitro, inhibiting secretion of IL-6, IL-8 and TNFα (*Mrowietz et al., 1992*; *Mrowietz et al., 1997*). Its effects on neutrophils were also shown in vitro, where dithranol leads to an increase in superoxide generation and a decrease in leukotriene production in neutrophils (*Kavanagh et al., 1996*; *Schröder, 1986*). Moreover, the potential receptor of dithranol remains undefined. Anti-psoriatic effects of other topical treatment options have recently been linked to modulation of the aryl hydrocarbon receptor (AhR) (*Smith et al., 2017*) and AhR may play a role in pathogenesis of psoriasis (*Di Meglio et al., 2014*). However, it appears that dithranol does not act via modulation of AhR, as we did not observe differences in the response of skin to dithranol comparing AhR knockout mice to controls (data not shown). There is need for

further studies investigating how dithranol exactly acts on the molecular level, which receptor it potentially binds to or inhibits or whether it acts through modulation of a specific transcription factor such as *NF-kB* or *STATs*. Another question is whether the effect of dithranol is specific compared to other topical treatments such as steroids and vitamin D3 analogues. There seem to be some overlapping mechanisms between dithranol and vitamin D3 analogues such as calcipotriol that has been shown to act on keratinocytes to repress the expression of IL-36α/γ, an effect mediated through keratinocytic vitamin D receptor (*Germán et al., 2019*). Moreover, similar to dithranol, calcipotriol decreased expression of AMPs such as ß-defensins in keratinocytes of psoriatic plaques (*Peric et al., 2009*). At the same time, calcipotriol normalized the proinflammatory cytokine milieu and decreased IL-17A, IL-17F and IL-8 transcript abundance in lesional psoriatic skin. Calcipotriol also directly targets Th17 cells (*Fujiyama et al., 2016*) and CD8+IL-17+ cells (*Dyring-Andersen et al., 2015*), whereas we found that dithranol only has delayed effects on T cells. Moreover, cathelicidin (LL37) expression was increased by calcipotriol (*Peric et al., 2009*), juxtaposing the results of dithranol treatment at least in the mouse-tail test of present study. That dithranol and vitamin D3 analogues may have similar basic mechanisms of action is also supported by the notion that depending on the concentration, both dithranol (*Nasimi et al., 2019*; *Ngwanya et al., 2017*) and calcipotriol (*El Taieb et al., 2019*; *Fullerton et al., 1998*; *Molinelli et al., 2020*) can be irritant to the skin and induce hair regrowth in alopecia areata. Compared to vitamin D3 analogues topical steroids have an even broader mechanisms of action in psoriasis linked to their anti-inflammatory, antiproliferative, vasoconstrictive (*Uva et al., 2012*) and immunomodulatory properties, in particular suppressing the IL-23/IL-17 axis, with IL-23 produced by dendritic cells/macrophages and IL-17 produced by Th17 cells/γδ T cells/innate lymphoid cells (*Germán et al., 2019*).

Another question that this study does not answer is, whether topical dithranol therapy has any effects on systemic psoriatic inflammation. However nowadays, treatment with dithranol is mainly administered in refractory, circumscribed psoriatic lesions in patients who do not have significant systemic inflammation how it may be otherwise the case in patients with moderate to severe forms of psoriasis.

Together our work opens up several avenues for novel topical (and potentially also systemic) treatment strategies in psoriasis. Not only targeting the IL-36 pathway, but also keratinocyte differentiation regulators (e.g. involucrin), keratinocyte-produced AMPs (ß-defensins like *DEFB4A*, *DEFB4B*, *DEFB103A*, S100 proteins like *S100A7, S100A12*), and neutrophils and their chemotactic factors (*CXCL5* and *CXCL8*) or members of the serpin family (*SERPINB7* and *SERPINB13*), are promising approaches. Such approaches may not only be helpful for chronic plaque-psoriasis, but also for pustular psoriasis, in which a vicious loop between AMPs such as cathelicidin (LL-37) and IL-36 signaling may drive psoriatic disease (*Benezeder and Wolf, 2019*; *Furue et al., 2018*; *Li et al., 2014*; *Madonna et al., 2019*; *Ngo et al., 2018*).

## Materials and methods

**Key resources table**

| Reagent type (species) or resource | Designation | Source or reference | Identifiers | Additional information |
|---|---|---|---|---|
| Antibody | anti-CD1a; mouse monoclonal | Immunotech, Beckman Coulter | Clone: O10 RRID:AB_10547704 | (undiluted) |
| Antibody | anti-CD3; mouse monoclonal | Novocastra, Leica Biosystems | Clone: PS1 RRID:AB_2847892 | (1:100) |
| Antibody | anti-CD4; mouse monoclonal | Novocastra, Leica Biosystems | Clone: 1F6 RRID:AB_563559 | (1:30) |
| Antibody | anti-CD8; mouse monoclonal | Dako, Agilent | Clone: C8/144b RRID:AB_2075537 | (1:25) |
| Antibody | anti-CK16; rabbit monoclonal | Abcam | Clone: EPR13504 RRID:AB_2847885 | (1:1000) |
| Antibody | anti-FoxP3; mouse monoclonal | Bio-Rad | Clone: 236A/E7 RRID:AB_2262813 | (1:100) |

*Continued on next page*

*Continued*

| Reagent type (species) or resource | Designation | Source or reference | Identifiers | Additional information |
|---|---|---|---|---|
| Antibody | anti-Ki-67; mouse monoclonal | Dako, Agilent | Clone: MIB-1 RRID:AB_2631211 | (1:50) |
| Antibody | anti-MPO; mouse monoclonal | Dako, Agilent | Clone: MPO-7 RRID:AB_578599 | (1:100) |
| Strain, strain background *Mus musculus* | BALB/c, wild-type | Charles River Laboratories | RRID:IMSR_CRL:028 Charles River Strain code#: 028 | |
| Strain, strain background *Mus musculus* | JunB$^{f/f}$ c-Jun$^{f/f}$ K5-Cre-ER$^{T}$ | PMID:16163348 | | Obtained from the laboratory of Maria Sibilia (Medical University of Vienna) |
| Commercial assay or kit | miRNeasy Mini Kit | Qiagen | Cat #: 217004 | |
| Commercial assay or kit | iScript Reverse Transcription Supermix | Bio-Rad | Cat #: 1708841 | |
| Commercial assay or kit | GoTag qPCR Master Mix | Promega | Cat #: A6001 | |
| Commercial assay or kit | Human GeneChip 2.0 ST arrays | Affymetrix, ThermoFisher Scientific | Cat #:902113 | |
| Commercial assay or kit | mouse Clariom S Assay | Affymetrix, ThermoFisher Scientific | Cat #:902919 | |
| Commercial assay or kit | GeneChip WT PLUS Reagent Kit | ThermoFisher Scientific | Cat #: 902280 | |
| Commercial assay or kit | GeneChip WT Terminal Labeling Kit | ThermoFisher Scientific | Cat #: 900671 | |
| Commercial assay or kit | GeneChip Hybridization, Wash and Stain Kit | ThermoFisher Scientific | Cat #: 900720 | |
| Commercial assay or kit | nCounter GX Custom codeset | NanoString Technologies | | Custom codeset (80 target genes, four reference genes |
| Chemical compound, drug | Aldara (Imiquimod) 5% cream | MEDA Pharmaceuticals | Cat #: 111981 | |
| Chemical compound, drug | Tamoxifen | Sigma-Aldrich | Cat #: T5648 | |
| Chemical compound, drug | Dithranol (1,8-Dihydroxy-9(10H)-anthracenone) | Gatt-Koller GmbH Pharmaceuticals | Cat #: 8069994 | Dissolved in vaseline and provided by the pharmacy of the Medical University of Graz, Austria |
| Software, algorithm | GraphPad Prism version 8 | GraphPad | RRID:SCR_002798 https://www.graphpad.com/scientific-software/prism/ | |
| Software, algorithm | Interacti Venn | PMID:25994840 | http://www.interactivenn.net/ | |
| Software, algorithm | Cytoscape | PMID:19237447 | RRID:SCR_003032 https://cytoscape.org/ | |
| Software, algorithm | Transcriptome Analysis Console (TAC) 4.0.2 | ThermoFisher Scientific | https://www.thermofisher.com/at/en/home/life-science/microarray-analysis/microarray-analysis-instruments-software-services/microarray-analysis-software/affymetrix-transcriptome-analysis-console-software.html | |
| Software, algorithm | Partek Genomics Suite version 6.6 | Partek Inc | RRID:SCR_011860 https://www.partek.com/partek-genomics-suite/ | |

*Continued on next page*

*Continued*

| Reagent type (species) or resource | Designation | Source or reference | Identifiers | Additional information |
|---|---|---|---|---|
| Software, algorithm | R Version 3.5.1 | The R Project for Statistical Computing | RRID:SCR_001905 https://www.r-project.org/ | |
| Software, algorithm | nSolver 2.5 Software | NanoString Technologies | https://www.nanostring.com/products/analysis-software/nsolver | |

## Patients

### Trial protocol and patient characteristics

For the clinical dithranol study, inclusion criteria were diagnosis of chronic plaque psoriasis, and age above 18 years. Exclusion criteria were intolerance of dithranol, autoimmune diseases, general poor health status, pregnancy and breast-feeding, topical treatment (steroids, vitamin D3-analogs and/or Vitamin A acid-derivates) within 2 weeks, and phototherapy within 4 weeks prior to study enrollment. None of the patients had received systemic treatment in the past prior to study enrollment. In total, 15 psoriasis patients (11 men, 4 women; median age 40.5 years, range 19.8–76.9 years) were enrolled. Mean duration of psoriasis had been 17.9 years (SD 11.5 years). Mean duration of dithranol treatment was 15.4 days (SD 3.6 days) and treatment was prematurely discontinued in two patients. Samples of normal skin from patients undergoing surgery for removal of benign skin lesions were available from 12 subjects (median age was 36.3 years, range 22.6–46.9 years) for control purposes. The samples were from lesion-adjacent, excised skin of patients who did not suffer from psoriasis, other inflammatory diseases or autoimmune diseases.

### Patient treatment

Dithranol ointment was prepared in the hospital pharmacy with 2% salicylic acid and white vaseline as base. It was administered to the patients daily in increasing concentrations; dosage was adjusted individually to the level of skin irritation. Concentration was usually increased every other day (starting from 0.1%, next 0.16%, 0.2%, 0.4%, 1% and finally 2%) and mean treatment duration was 15.4 days.

### Marker lesions and scoring

At each of the four visits (see *Figure 1*), i) Psoriasis Area and Severity index (PASI), ii) local psoriasis severity index (PSI) and erythema score of marker lesions and iii) perilesional erythema score were assessed. PSI was composed by rating of erythema, induration and scaling, each on a scale from 0 to 4, resulting in a maximum score of 12. Erythema score was determined by rating intensity of lesional erythema on a scale of 0–4 (0 = none, 1 = mild, 2 = moderate, 3 = severe, 4 = very severe erythema). For perilesional erythema score, intensity of perilesional erythema was rated on a scale of 0–3 (0 = none, 1 = mild, 2 = moderate, 3 = severe perilesional erythema). Per patient, four marker lesions of similar morphology and size and, if possible, from the same body regions or four marker areas (each 5 cm in diameter) of one or more larger psoriatic lesions were defined at study entry and then scored, treated with dithranol and later biopsy-sampled at certain timepoints.

### Patient tissue sampling

Biopsy samples were taken from the psoriasis patients before (day 0), during early treatment at first strong perilesional inflammation between day 4 and 9 (with most biopsies taken at day 6) at end of treatment (approximately after 2–3 weeks) and at a follow-up visit (4–6 weeks after end of therapy). Per patient, a total of five biopsy samples were taken on four study days by means of a punch cylinder (up to 5 mm) under local anesthesia. At the first visit before starting therapy, one biopsy sample was taken additionally from adjacent non-lesional skin, with a distance of at least 5 cm from the edge of psoriatic skin. One part of each biopsy was fixed in 4% neutral-buffered paraformaldehyde and used for histology and immunohistochemistry. The other part was stored in RNAlater solution (Invitrogen, California, USA) at −80°C until RNA extraction for further analysis.

## Histology and immunohistochemistry

### Analysis of HE stained sections

Human and murine samples fixed with paraformaldehyde were processed routinely, cut in 4 μm sections and stained with hematoxylin and eosin (HE). Five randomly selected fields per slide were investigated for histological analysis. Thickness of epidermis was measured from basal layer to stratum corneum using an Olympus BX41 microscope (Olympus Life Science Solutions, Hamburg, Germany), cellSens software (Olympus Life Science Solutions) and 20x magnification. Semi-quantitative scoring (0 = none, 0.5 = none/low, 1 = low, 1.5 = low/moderate, 2 = moderate, 2.5 = moderate/high, 3 = high density of infiltrate) was performed at five randomly selected locations per slide and at 20x magnification.

In the mouse-tail model, degree of orthokeratosis was analyzed as described by *Bosman et al., 1992* . In brief, five randomly selected scales per sample were examined and the length of the granular layer (A) as well as the total length of the scale (B) were measured using cellSens software (Olympus-lifescience, Hamburg, Germany) and 20x magnification. The proportion of (A/B) x 100 depicts the percentage of orthokeratosis per scale.

### Immunohistochemistry stainings and analysis

Antigen retrieval was performed using either EDTA-buffer (CD1a, CD3, CD4, CK16, CD8), citrate-buffer (FoxP3, Ki-67) or trypsin (MPO). Primary antibodies used were: anti-human CD1a (mouse monoclonal, clone O10, undiluted; Immunotech, Beckman Coulter, Praque, Czech Republic), anti-human CD3 (mouse monoclonal, clone PS1, dilution 1:100; Novocastra, Leica Biosystems, Mannheim, Germany), anti-human CD4 (mouse monoclonal, clone 1F6, dilution 1:30; Leica Biosystems), anti-human CD8 (mouse monoclonal, clone C8/144b, dilution 1:25, Dako Omnis, Agilent, Santa Clara, CA, USA), anti-human CK16 (rabbit monoclonal, clone EPR13504, dilution 1:1000, Abcam, Cambridge, UK), anti-human FoxP3 (clone 236A/E7, AbD Serotec, Bio-Rad, Hercules, CA, USA), anti-human Ki-67 (mouse monoclonal, clone MIB-1, dilution 1:50, Dako Omnis, Agilent) and anti-human MPO (mouse monoclonal, clone MPO-7, dilution 1:100, Dako Omnis, Agilent). Stainings were performed using the Dako REALTM Detection System, Peroxidase/AEC, rabbit/mouse (Dako, Agilent) on the Dako Autostainer Link 48 (Dako, Agilent) according to the manufacturer's instructions. For quantification of CD1a, CD3, CD4, FoxP3, CD8, and MPO staining, all positively stained cells with visible nucleus in five randomly selected fields (separately for epidermis and dermis) per slide were counted and results were averaged to obtain mean cell counts. To quantify Ki-67 and CK16 staining, area of positive staining was divided by epidermal area as follows: one representative image per slide was taken on an Olympus BX41 microscope (Olympus Life Science Solutions, Hamburg, Germany) at 10x magnification using cellSens software (Olympus Life Science Solutions). TIFF images were imported into ImageJ program and pixels were converted to μm. Using the polygon sections tool, the outline of epidermis was framed, and the total area was measured. In addition, total area of particles (positively stained cells) within the epidermal area was assessed.

## Gene expression analyses

### RNA extraction

Total RNA was extracted from frozen biopsies of psoriasis patients and control subjects. To facilitate homogenization, tissues were cut in 20 μm sections using a cryomicrotome and collected in pre-cooled MagNA Lyser Green Beads tubes (Roche, Basel, Switzerland) and disruption of tissue was performed on the MagNA Lyser Instrument (Roche, Basel, Switzerland). After efficient homogenization, total RNA was extracted using the miRNeasy Mini Kit (Qiagen, Hilden, Germany), according to the manufacturer's instructions. To ensure complete DNA removal, on-column DNase digestion was performed and RNA was eluted in 15–20 μl RNase-free water. Mouse tissues were handled in the same way except that sufficient homogenization of samples was obtained without using a cryomicrotome.

### Microarray and pathway analysis

Isolated RNA was quality checked on a Bioanalyzer BA2100 (Agilent; Foster City, CA) using the RNA 6000 Nano LabChip according to manufacturer's instructions and samples with RNA integrity numbers (RIN) between 5 to 8 were considered for analysis using Human GeneChip 2.0 ST arrays

(Affymetrix, ThermoFisher Scientific, Waltham, MA, USA; Cat.No.: 902113) for the human samples and mouse Clariom S Assay (Affymetrix, ThermoFisher Scientific, Waltham, MA, USA; Cat No. 902919) for the mouse samples. For first and second strand cDNA synthesis 500 ng total RNA were used in the GeneChip WT PLUS Reagent Kit (ThermoFisher) according to manufacturer's instructions. Fragmentation and terminal labelling was performed using the GeneChipTM WT Terminal Labeling Kit (ThermoFisher) and hybridization, wash and stain of arrays was performed on a Gene-Chip Fluidics 450 station using the GeneChip Hybridization, Wash and Stain Kit (ThermoFisher) according to manufacturer's instructions. Arrays were scanned in a GeneChip GCS300 7G Scanner and analysed with the Affymetrix Expression Console Software 1.3.1 (ThermoFisher) for the human array and Transcriptome Analysis Console (TAC) 4.0.2 for the mouse arrays. Samples were processed at the Core Facility Molecular Biology at the Centre of Medical Research at the Medical University of Graz, Austria. Pre-processing of CEL-files for the human arrays was performed with Partek Genomics Suite version 6.6 (Partek Inc, St Louis, MO, USA) using the robust multi-chip analysis (RMA) algorithm, which includes background adjustment, quantile normalisation and median polished probe summarisation. For statistical analysis, paired-sample t-tests were used and genes with p<0.05 and a fold change of at least 1.5 were considered to be significantly deregulated. All microarray data has been deposited at the public repository Gene Expression Omnibus (GEO) (http://www.ncbi.nlm.nih.gov/geo/) with accession numbers GSE145126 and GSE145127.

For analysis of histological responders compared to non-responders, as well as mouse arrays, pre-processing and RMA normalization was done with RStudio Version 1.2.1335 (R Version 3.5.1) with MicroArrayPipeline v1.0 Shiny app based on limma Bioconductor package for differential expression analysis and genes with p<0.05 and a fold change of at least 1.5 were considered to be significantly deregulated. Mouse and human DEGs were tested for potential overlap. Probeset IDs of early (day 6) as well as late (week 2–3) DEGs of human microarray were matched with DEGs of mouse microarray using NetAffxTM Expression Array Comparison Tool.

## Nanostring nCounter analysis and microarray verification

For Nanostring analysis, a nCounter GX Custom codeset (80 target genes, four reference genes, see *Supplementary file 2*) was designed to verify microarray results of selected DEGs. Total RNA (150 ng) with RIN values between 4.7 and 9 was used and samples were processed according to supplier's instructions (NanoString Technologies, Seattle, WA USA) and scanned on a nCounter Digital Analyzer (NanoString Technologies). RCC files were used for data analysis. Raw data pre-processing and normalization was performed using nSolver 2.5 Software (NanoString Technologies) according to standard procedures (background subtraction, positive and negative controls normalization). Gene counts were then normalized to the geometric mean of the reference genes. Normalized data was uploaded to Partek Genomic Suite Software v6.6 (Partek Inc, St Louis, MO, USA) and paired t-test was used for statistical analysis. Nanostring and microarray fold change values of selected target genes were log-transformed and the two platforms were compared by Pearson correlation of each gene across samples.

## Gene ontology enrichment analysis

For all comparisons genes with a p-value<0.05 and FC ± 1.5 were assigned to GO enrichment analysis using Cytoscape software ver.3.5.1 (Cytoscape Consortium, NY, USA www.cytoscape.org; *Bindea et al., 2009*). Gene identifiers were loaded into Cytoscape software and ClueGO analysis was used to identify significantly overrepresented GO terms and associated genes. P-values (significance level <0.05) were adjusted using Bonferroni step-down corrections.

## RT-qPCR

Per sample 2 µg of RNA was reverse transcribed into cDNA using iScript Reverse Transcription Supermix (Bio-Rad, Hercules, CA, USA). Relative gene expression was determined using GoTag qPCR Master Mix (Promega, Mannheim, Germany) on a CFX96 Touch Real-Time PCR Detection System (Bio-Rad). The following cycling conditions were used: Hot-start activation (95°C, 2 min), denaturation for 40 cycles (95°C, 15 s) and annealing/extension (60°C, 60 s). Melting curve analysis was done to confirm amplification specificity. For each sample, qPCRs were run in triplicates. Cycle

thresholds (Ct) were determined and relative mRNA expression to *Ubc or Ywhaz* (reference genes) were calculated using the ΔCt method. Primer sequences are listed in *Supplementary file 4*.

## Animals

### Mouse strains and housing

6–9 week-old mice were kept with food and water ad libitum in the conventional animal facility at the Centre for Medical Research, Medical University of Graz or at the Medical University of Vienna, Austria. During experiments, all mice were monitored closely to ensure sufficient health status. BALB/c mice were purchased from Charles River (Sulzfeld, Germany). c-Jun/JunB knockout mice (*Zenz et al., 2005*) were bred and maintained in the facilities of the Medical University of Vienna. All animal experiments were in accordance with institutional policies and federal guidelines.

### Therapeutic agents used in mice

For all animal experiments, dithranol in different concentrations (dissolved in vaseline) and vehicle (vaseline cream only) was provided by the pharmacy of the Medical University of Graz, Austria. Aldara (IMQ) 5% cream (MEDA Pharmaceuticals, Vienna, Austria) and tamoxifen (Sigma-Aldrich, Missouri, USA) were purchased.

### Imiquimod model

24 hr before starting an experiment, dorsal skin of BALB/c mice was shaved carefully. To induce psoriasis-like dermatitis, imiquimod (IMQ) cream was applied daily for five consecutive days on dorsal skin (approximately 40 mg) and right ear (approximately 20 mg). A daily topical dose of 62.5 mg of IMQ cream was not exceeded; translating into 3.125 mg of the active compound. Mice received IMQ cream in the morning and dithranol treatment in the afternoon, with a time gap of 8 hr. Dithranol was applied topically on dorsal skin (40 mg) and right ear (20 mg) and concentrations were increased every other day (0.01% on day 1–2, 0.03% on day 3–4% and 0.1% on day 5–6). Control mice were treated similarly with Vaseline cream. Double skin fold of dorsal skin and ear thickness was measured daily in triplicates before any application using a micrometer (Mitutoyo, Kanagawa, Japan). 24 hr after the last topical treatment, mice were sacrificed and approximately 1 cm$^2$ of dorsal skin, both ears (treated and untreated as control) were collected.

### c-Jun/JunB knockout mouse model

Mice carrying floxed alleles for the JunB and c-Jun locus and expressing the K5-CreERT transgene (mixed background) received consecutive i.p. injections of tamoxifen (1 mg/day) for a period of 5 days, leading to deletion of both genes in the epidermis by inducible Cre-recombinase activity (*Schonthaler et al., 2009*; *Zenz et al., 2005*). One week after the last injection, psoriasis-like lesions on the ears were treated daily with dithranol in series of increasing concentrations (0.01% on day 1–2, 0.03% on day 3–4% and 0.1% on day 5–6 or 0.1% on day 1–2, 0.3% on day 3–4% and 1% on day 5–6) as depicted in *Figure 3A*. Control mice were treated similarly with vehicle cream. Ear thickness was measured daily by micrometer before any topical application and 24 hr after the last topical treatment mice were sacrificed and ears were collected.

### Mouse tail test

For the mouse-tail model, 40 mg dithranol 1% was applied daily for 14 days to the surface of the proximal part of tails (approx. 2 cm in length starting 1 cm from the body). 24 hr after the last application, mice were sacrificed, and treated parts of tail skin were removed from the underlying cartilage. The mouse-tail test for psoriasis is a traditional model to quantify the effect of topical antipsoriatics on keratinocyte differentiation by measuring degree of orthokeratosis versus parakeratosis (*Bosman et al., 1992*; *Sebök et al., 2000*; *Wu et al., 2015*).

## Statistical analyses

Statistical analyses for human and murine microarrays were performed as described in the specific sections. All other statistical analyses were determined using GraphPad Prism version 8 (GraphPad software, California, USA). Normality was determined by Shapiro-Wilk test and differences between two groups were assessed by Mann Whitney test, Wilcoxon test or T-test (paired or unpaired) as

appropriate. For multiple comparisons, One-way ANOVA with Dunnett's test or Tukey's test was used for parametric data and Kruskal Wallis test with Dunn's test was used for nonparametrical data as indicated in the specific sections. Significance was set at a p-value of $\leq$0.05. Each animal experiment was performed at least twice. For correlation analysis of clinical scores, as well as comparison of microarray and nanostring ratios, Pearson or Spearman correlation was used as indicated.

### Study approval

A clinical study (Clinical Trials.gov no. NCT02752672) in psoriatic patients treated with dithranol was completed in cooperation with the Department of Dermatology, Klagenfurt State Hospital. Clinical trial procedures were approved by the ethics committee of the federal state of Carinthia, Austria (protocol number A23/15) and all participants gave written informed consent in accordance with the principles of the Declaration of Helsinki. All mouse experiments were approved by the Austrian Government, Federal Ministry for Science and Research (protocol numbers BMWF-66-010/0032-11/3b/ 2018, 66.009/0200-WF/II/3b/2014) and animal experiments performed in Vienna were additionally approved by the Animal Experimental Ethics Committee of the Medical University of Vienna.

## Acknowledgements

The authors would like to thank all patients who made this work possible and Gerlinde Mayer andUlrike Schmidbauer, for technical support and Karin Wagner and Ingeborg Klymiuk for support in Microarray and Nanostring analyses, all Medical University of Graz; Martina Hammer, Medical University of Vienna, for support in animal experimentation; Ahmed Gehad and Rachael Clark for support in animal experimentation and scientific advice, both at Brigham and Women's Hospital, Harvard Medical School; and Honnovara Ananthaswamy, Houston, TX, for critical reading of the manuscript. The authors would also like to thank Kathrin Eller and Herbert Strobl, who provided valuable guidance as members of the thesis committee of TB. This work was supported by the Austrian Science Fund FWF (W1241) and the Medical University of Graz through the Ph.D. Program Molecular Fundamentals of Inflammation (DK-MOLIN) to PW; by grants from the Austrian Science Fund (FWF, W1212), the WWTF-project LS16-025, the European Research Council (ERC) Advanced grant (ERC-2015-AdG TNT-Tumors 694883) and the European Union's Horizon 2020 research and innovation program under the Marie Skłodowska-Curie grant agreement No. 766214 (Meta-Can) to MS. TB was supported by the Austrian Marshall Plan Foundation and the Austrian Society for Dermatology and Venereology (Klaus Wolff fellowship). CP was supported by the Elisabeth Hirschheiter research grant. PW was supported by a Visiting Scholar Award of the Human Skin Disease Resource Center (HSDRC) Grant Program (NIH/NIAMS P30AR069625), Department of Dermatology, Brigham and Women's Hospital, Harvard Medical School.

## Additional information

### Funding

| Funder | Grant reference number | Author |
|---|---|---|
| Austrian Science Fund | W1241 | Peter Wolf |
| Medical University of Graz | PhD Program Molecular Fundamentals of Inflammation (DK-MOLIN) | Peter Wolf |
| Austrian Science Fund | W1212 | Maria Sibilia |
| Vienna Science and Technology Fund | LS16-025 | Maria Sibilia |
| H2020 European Research Council | ERC-2015-AdG TNT-Tumors 694883 | Maria Sibilia |
| Horizon 2020 Framework Programme | 766214 | Maria Sibilia |
| Marshallplan-Jubiläumsstiftung | | Theresa Benezeder |

| Austrian Society for Dermatology and Venereology | | Theresa Benezeder |
|---|---|---|
| Elisabeth Hirschheiter research grant | | Clemens Painsi |
| National Institute of Arthritis and Musculoskeletal and Skin Diseases | P30AR069625 | Peter Wolf |

The funders had no role in study design, data collection and interpretation, or the decision to submit the work for publication.

### Author contributions

Theresa Benezeder, Conceptualization, Data curation, Formal analysis, Validation, Investigation, Visualization, Methodology, Writing - original draft, Project administration; Clemens Painsi, Conceptualization, Data curation, Investigation, Project administration; VijayKumar Patra, Visualization, Methodology; Saptaswa Dey, Methodology; Martin Holcmann, Resources, Methodology; Bernhard Lange-Asschenfeldt, Resources, Project administration; Maria Sibilia, Resources, Writing - review and editing; Peter Wolf, Conceptualization, Supervision, Funding acquisition, Writing - review and editing

### Author ORCIDs

Theresa Benezeder (ID) https://orcid.org/0000-0001-6218-2792
VijayKumar Patra (ID) https://orcid.org/0000-0002-7161-7767
Saptaswa Dey (ID) https://orcid.org/0000-0001-7532-7858
Peter Wolf (ID) https://orcid.org/0000-0001-7777-9444

### Ethics

Clinical trial registration Clinical Trials.gov no. NCT02752672.

Human subjects: A clinical study in psoriatic patients treated with dithranol was completed in cooperation with the Department of Dermatology, Klagenfurt State Hospital. Clinical trial procedures were approved by the ethics committee of the federal state of Carinthia, Austria (protocol number A23/15) and all participants gave written informed consent in accordance with the principles of the Declaration of Helsinki.

Animal experimentation: All mouse experiments were approved by the Austrian Government, Federal Ministry for Science and Research (protocol numbers BMWF-66-010/0032-11/3b/2018, 66.009/0200-WF/II/3b/2014) and animal experiments performed in Vienna were additionally approved by the Animal Experimental Ethics Committee of the Medical University of Vienna.

### Decision letter and Author response

Decision letter https://doi.org/10.7554/eLife.56991.sa1
Author response https://doi.org/10.7554/eLife.56991.sa2

## Additional files

### Supplementary files

• Supplementary file 1. Top 45 upregulated genes in lesional psoriatic skin compared to non-lesional skin at baseline from 15 patients with psoriasis (p<0.05, fold change >1.5).

• Supplementary file 2. Verification of microarray results of dithranol-treated skin samples of psoriasis patients with Nanostring analysis (nCounter GX Custom codeset) of 80 target genes and four reference genes. FC, fold change

• Supplementary file 3. Top 20 significantly enriched pathways as determined by Gene Ontology (GO) enrichment analysis in histological responders compared to non-responders in the psoriasis patient cohort (GO was done using Cytoscape software; [*Bindea et al., 2009*; *Shannon et al., 2003*] a.o. = among others).

- Supplementary file 4. qPCR Primer sequences and corresponding annealing temperatures.
- Transparent reporting form

## Data availability

All microarray data has been deposited at the public repository Gene Expression Omnibus (GEO) (http://www.ncbi.nlm.nih.gov/geo/) with accession numbers GSE145126 and GSE145127.

The following datasets were generated:

| Author(s) | Year | Dataset title | Dataset URL | Database and Identifier |
|---|---|---|---|---|
| Benezeder T, Painsi C, Patra V, Dey S, Holcmann M, Lange-Asschenfeldt B, Sibilia M, Wolf P | 2020 | Microarray analysis of c-Jun/JunB knockout mice treated with dithranol | https://www.ncbi.nlm.nih.gov/geo/query/acc.cgi?acc=GSE145126 | NCBI Gene Expression Omnibus, GSE145126 |
| Benezeder T, Painsi C, Patra V, Dey S, Holcmann M, Lange-Asschenfeldt B, Sibilia M, Wolf P | 2020 | Microarray analysis of dithranol-treated psoriasis | https://www.ncbi.nlm.nih.gov/geo/query/acc.cgi?acc=GSE145127 | NCBI Gene Expression Omnibus, GSE145127 |

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
