## [Decision Letter]

**Acceptance summary:**

The role of keratinocytes in the pathogenesis of psoriasis has gained a lot of interest recently and your work supplements the new insights from the therapeutic point of view. Dithranol by itself may still be a therapeutic niche, but the data regarding the mode of action can stimulate drug development along these lines.

**Decision letter after peer review:**

Thank you for submitting your article "Dithranol targets keratinocytes, their crosstalk with neutrophils and inhibits the IL-36 inflammatory loop in psoriasis" for consideration by *eLife*. Your article has been reviewed by three peer reviewers, one of whom is a member of our Board of Reviewing Editors, and the evaluation has been overseen by Jos van der Meer as the Senior Editor. The following individual involved in review of your submission has agreed to reveal their identity: Claus Johansen (Reviewer #3).

The reviewers have discussed the reviews with one another and the Reviewing Editor has drafted this decision to help you prepare a revised submission.

As the editors have judged that your manuscript is of interest, but as described below that additional work is required before it is published, we would like to draw your attention to changes in our revision policy that we have made in response to COVID-19 (https://elifesciences.org/articles/57162). First, because many researchers have temporarily lost access to the labs, we will give authors as much time as they need to submit revised manuscripts. We are also offering, if you choose, to post the manuscript to bioRxiv (if it is not already there) along with this decision letter and a formal designation that the manuscript is "in revision at *eLife*". Please let us know if you would like to pursue this option. (If your work is more suitable for medRxiv, you will need to post the preprint yourself, as the mechanisms for us to do so are still in development.)

Summary:

Topic of interest but in major parts confirmatory to already published work and in other parts not conclusive. Please provide clarity about the data on AMPs in this manuscript and in the manuscript submitted to the other journal.

Essential revisions:

Statistical analysis (see reviewer 2). Interpretation of data (specific for dithranol or bystander effects (see reviewers 1 and 2). Interpretation of data regarding the mouse models used as there seem a number of inconsistencies (see reviewers 2 and 3).

Interpretation of results in the light of the critique of all reviewers.

Reviewer #1:

The manuscript by Benezeder et al. addresses an interesting topic. The role of keratinocytes for psoriasis pathogenesis has been neglected for many years and the work presented here re-focuses on keratinocyte-mediated effects.

Interestingly, the authors used dithranol as a model, which is challenging since this topical compound shows beneficial therapeutic effects but also induces dose- and individual-depended skin irritation. Although the methods employed are well established and state-of-the-art there are some points that need clarification.

• Psoriasis is viewed as systemic disorder and to judge the overall beneficial effect of any topical therapy including dithranol it should be shown whether there are effects on systemic inflammation (e.g. vascular inflammation). The authors should mention this as limitation of their approach.

• The most commonly used topical therapies are corticosteroids alone or in a fixed combination with a vitamin D analogue. The authors are encouraged to use such treatment as a control in order to prove that the observed effects are specific for dithranol.

• The irritant potential of dithranol is helpful in cases of otherwise refractory alopecia areata. The authors should briefly mention this and discuss if their data may explain the effect as an outlook for further investigation.

The reviewer is puzzled by the fact that he was asked to review a manuscript of the same group with the same first author with significant similarity to data that are described in this manuscript for another journal particularly about the effect of dithranol on antimicrobial peptides.

The authors must make clear if there is an overlap of data and which of the data presented in the other manuscript are also contained in this one.

Reviewer #2:

Benezeder et al. present a paper on dithranol action on keratinocytes in psoriasis. The authors provide data on clinical outcome, reduction of epidermal thickness, cell proliferation and keratin 16 expression. All changes were observed at week 2-3 of treatment. By microarray analysis on day 6 and week 2-3 differentially expressed genes (DEGs) were identified, among these genes such encoding keratinocyte differentiation, skin barrier function and antimicrobial peptides (AMPs), but also IL-17 or IL-36. In addition, the authors used dithranol treatment in a c-Jun/JunB knockout mouse model. Again, microarray analysis was used to follow effects. The authors focused on genes belonging to keratinocyte function. Microarray data generated from the c-Jun/JunB knockout mouse model was compared to human microarray data. Dithranol was also applied to mouse tails. Here, they found increased thickening and orthokeratosis, increase in keratin 16 expression and AMPs. Finally, they used the imiquimod psoriasis-like mouse model where dithranol increased skin thickening. The authors conclude that the effects of dithranol are mainly mediated by keratinocytes and IL-36.

Although this is an interesting study, there are some major caveats and conflicting results, which make it difficult to accept this paper.

1) The main messages of this study – decrease of psoriatic epidermal thickening, reduction of Ki67 positivity, reduction of keratin 16 expression and AMPs by dithranol have been shown by previous reports from human skin (van de Kerkhof et al., 1996 and 2002; Eberle et al., 2017).

2) There is plenty of evidence demonstrating that psoriasis is an IL-17-driven immune disease. All treatments with a rapid clinical response will find a decrease in epidermal thickening, a normalization of keratinocyte differentiation and a decrease in immune cell infiltration as well as cytokine expression. From this perspective, one cannot conclude that the one or other treatment will work through keratinocytes and not influence the immune system. The changes in keratinocytes as observed in Figure 2 (epidermal thickening, Ki67 expression) seem to be significant on week 2 to 3. A similar pattern with a decrease in epidermal thickening, keratin 16 and Ki67 expression is visible at week 2 of treatment when using monoclonal antibodies e.g. directed against IL-17. Similarly, antimicrobial peptides are regulated at early time points by immunobiologics (several well-performed studies on this topic are published by Krueger and colleagues in JACI).

3) The mouse model used (c-Jun/JunB knockout) is somehow artificial and does not require T cells for developing epidermal changes. Therefore, the findings – although of interest – cannot be translated to the human situation.

4) The Major concern is that most of the findings are descriptive. The authors performed many microarray analyses but studies on functional mechanisms confirming their hypothesis are missing.

5) The increase in epidermal thickening, the changes in keratin 16 and AMPs found in the mouse tail experiment with dithranol contradict the results found in human psoriasis (Figure 4).

6) IL-36 cytokines are mainly expressed by keratinocytes (Wenzel et al., J Invest Dermatol 2015), but the authors describe a decrease at week 2 to 3. If dithranol targets primarily keratinocytes and inhibits IL-36, the effect should be visible at earlier time points.

Reviewer #3:

The manuscript by Benezeder et al. investigates the mode of action by which dithranol mediates its anti-psoriatic effects. Although dithranol has been used as topical treatment for psoriasis for many years the mechanism by which dithranol works is not that well described and therefore a better understanding of the mode of action of dithranol is needed and could pave the way for other topical drugs for psoriasis.

The manuscript is interesting and well-written and the data are convincing.

I have the following comments:

1) Although the data in the manuscript clearly shows that the early mode of action of dithranol in psoriasis treatment is on keratinocyte-derived molecules and subsequently neutrophil infiltration to the skin the manuscript does not reveal the molecular mechanism behind this. Does dithranol use a membrane bound receptor, an intracellular receptor or does it inhibit specific transcription factors (e.g. NF-κB, STAT or IkBzeta)? Have the authors investigated this or is there anything in the literature about this. The authors should discuss this more in the manuscript.

2) Surprisingly, dithranol does not have an anti-psoriatic effect in the imiquimod-induced psoriasis model. In contrast, dithranol has an anti-psoriatic effect in the c-Jun/JunB mouse model. Were the mouse strains used in the two models the same? Were the mice similar in age and did the authors use females or males and was this the same in the two models? The authors write that dithranol rather worsened the psoriatic lesions in the IMQ model. And interestingly the authors write that "Dithranol treatment of healthy murine skin led to similar effects". Could this be the same effect as has been described in perilesional skin? Is the inflammatory effect of dithranol on perilesional skin known? If so the authors could examine if this is also what is seen in skin of IMQ-treated mice. These issues could be discussed in more detail.

3) In the psoriasis patients treated with dithranol biopsies were taken from lesional skin over time. Did the authors take biopsies from the same "target" lesion throughout the study?

4) In Table 1 the authors should in the legend include that the numbers in the table are based on data from 15 psoriatic patients.

---

## [Author Response]

Summary:Topic of interest but in major parts confirmatory to already published work and in other parts not conclusive. Please provide clarity about the data on AMPs in this manuscript and in the manuscript submitted to the other journal.

We analyzed biopsy samples from 15 psoriasis patients at several time points, i.e., before, during, after dithranol therapy and at a follow-up visits and from a well-matched control group (in a total more than 70 human biopsies) and used three mouse models (IMQ model, c-Jun/JunB model and mouse tail test) to substantiate our findings on the effects of dithranol in diseased psoriatic skin. We are first by using genome-wide expression analysis to investigate the effects of dithranol and by doing so we were able to thoroughly dissect the mechanisms of dithranol in diseased psoriatic skin. As pointed out, psoriatic response to dithranol was characterized by a rapid decrease in expression of keratinocyte differentiation regulators (e.g. involucrin, *SERPINB7* and *SERPINB13*), anti-microbial peptides (e.g. ß-defensins like *DEFB4A, DEFB4B, DEFB103A*, S100 proteins like *S100A7, S100A12*), chemotactic factors for neutrophils (e.g. *CXCL5*, *CXCL8*) and neutrophilic infiltration and was followed with much delay by reduction in T cell infiltration. These findings are novel and haven't been reported previously.

We understand the reviewer’s concern that we recently submitted another manuscript about dithranol’s effects to a different journal. However, in the latter manuscript (Letter to the Editor) we report on dithranol's irritant effect (possibly playing a role in the therapy of alopecia areata) on healthy skin exclusively using healthy murine skin as well as xenografted healthy human skin. We want to point out that there is no overlap at all between the data and conclusions of the two manuscripts. However, we are happy to provide this letter on a confidential basis to *eLife* if the editors wish so.

Essential revisions:Statistical analysis (see reviewer 2). Interpretation of data (specific for dithranol or bystander effects (see reviewers 1 and 2). Interpretation of data regarding the mouse models used as there seem a number of inconsistencies (see reviewers 2 and 3).Interpretation of results in the light of the critique of all reviewers.

We have responded to all these comments appropriately (see below), including a revision of statistics.

Reviewer #1:The manuscript by Benezeder et al. addresses an interesting topic. The role of keratinocytes for psoriasis pathogenesis has been neglected for many years and the work presented here re-focuses on keratinocyte-mediated effects. Interestingly, the authors used dithranol as a model, which is challenging since this topical compound shows beneficial therapeutic effects but also induces dose- and individual-depended skin irritation.Although the methods employed are well established and state-of-the-art there are some points that need clarification.• Psoriasis is viewed as systemic disorder and to judge the overall beneficial effect of any topical therapy including dithranol it should be shown whether there are effects on systemic inflammation (e.g. vascular inflammation). The authors should mention this as limitation of their approach.

We agree with the reviewer and have mentioned this limitation in the tenth paragraph of the Discussion.

• The most commonly used topical therapies are corticosteroids alone or in a fixed combination with a vitamin D analogue. The authors are encouraged to use such treatment as a control in order to prove that the observed effects are specific for dithranol.

Indeed, some downstream effects of dithranol seem to be overlapping with that of vitamin D3 analogues (more than with that of topical steroids), whereas other effects such as keratinocyte-neutrophil crosstalk are differential and seem to be specific for dithranol. We discuss this now in the Discussion and provide a couple of references on papers investigating the molecular effects of calcipotriol and topical steroids on human psoriasis.

• The irritant potential of dithranol is helpful in cases of otherwise refractory alopecia areata. The authors should briefly mention this and discuss if their data may explain the effect as an outlook for further investigation.

We entirely agree with the reviewer on the effect of dithranol in alopecia areata (see also comment above to the Editor) and have added a respective sentence to the eighth paragraph of the Discussion.

The reviewer is puzzled by the fact that he was asked to review a manuscript of the same group with the same first author with significant similarity to data that are described in this manuscript for another journal particularly about the effect of dithranol on antimicrobial peptides.The authors must make clear if there is an overlap of data and which of the data presented in the other manuscript are also contained in this one.

Please see response above to the Editor's comments.

Reviewer #2:[…] Although this is an interesting study, there are some major caveats and conflicting results, which make it difficult to accept this paper.1) The main messages of this study – decrease of psoriatic epidermal thickening, reduction of Ki67 positivity, reduction of keratin 16 expression and AMPs by dithranol have been shown by previous reports from human skin (van de Kerkhof et al., 1996 and 2002; Eberle et al., 2017).

Please see response above to the Editor's comments. We would like to stress that we discuss the results of those previous studies by van de Kerkhof et al. and Eberle et al., 2017 now in more detail in our paper (Discussion, second paragraph). In addition, we wish to point out that we are the first to show dithranol’s effect on a transcriptional level in a large patient cohort with multiple time points of biopsy sampling and in two psoriasis mouse models. Moreover, our main conclusion is that inhibition of keratinocyte-neutrophil crosstalk as well as IL-36 pathway is crucial for its anti-psoriatic effect. These findings are entirely novel.

2) There is plenty of evidence demonstrating that psoriasis is an IL-17-driven immune disease. All treatments with a rapid clinical response will find a decrease in epidermal thickening, a normalization of keratinocyte differentiation and a decrease in immune cell infiltration as well as cytokine expression. From this perspective, one cannot conclude that the one or other treatment will work through keratinocytes and not influence the immune system. The changes in keratinocytes as observed in Figure 2 (epidermal thickening, Ki67 expression) seem to be significant on week 2 to 3. A similar pattern with a decrease in epidermal thickening, keratin 16 and Ki67 expression is visible at week 2 of treatment when using monoclonal antibodies e.g. directed against IL-17. Similarly, antimicrobial peptides are regulated at early time points by immunobiologics (several well-performed studies on this topic are published by Krueger and colleagues in JACI).

We agree with the reviewer, however, our data clearly indicate that dithranol treatment first acts on keratinocytes and later by delayed direct or indirect effects on the immune system. Indeed, certain biologics even of the latest generation have similar effects on the molecular level, which is fascinating, that a traditional topical agent like dithranol can show similar effects as a systemic antibody therapy. We believe that dithranol’s primary effect on keratinocytes is crucial for response to treatment, considering that differentially regulated genes in histological responders compared to non-responders belonged to pathways like keratinocyte differentiation, cornification and keratin filament formation (as mentioned in the third paragraph of the Discussion and Supplementary file 3).

We thank the reviewer for hinting us to the work of Krueger et al., 2012 and 2019). We have slightly modified the last sentence of the Results section and added those references in the seventh paragraph of the Discussion section, together with a couple of respective sentences.

3) The mouse model used (c-Jun/JunB knockout) is somehow artificial and does not require T cells for developing epidermal changes. Therefore, the findings – although of interest – cannot be translated to the human situation.

We entirely agree that findings from mouse models cannot be directly translated to the human situation. However, in addition to the imiquimod model, a solely immunologically mediated psoriasis model, we specifically have chosen the c-Jun/JunB model that is based on keratinocytes rather than on the immune system. We had mentioned this in the manuscript but to make our purpose clearer, we have now modified the wording (Discussion, first paragraph): "The therapeutic importance of reduction of IL-1 family members in human skin was substantiated by results generated in the keratinocyte-based c-Jun/JunB mouse psoriasis model."

4) The Major concern is that most of the findings are descriptive. The authors performed many microarray analyses but studies on functional mechanisms confirming their hypothesis are missing.

We would have loved to substantiate our findings on the functional level and have mentioned this shortage of the paper "What our work does not answer, is how dithranol exactly acts at the molecular level. Cell culture studies have shown that dithranol targets mitochondria.…". Anti-psoriatic effects of other topical treatment options have recently been linked to modulation of AhR (Smith et al., 2017) and AhR may play a role in pathogenesis of psoriasis (DiMeglio et al., 2014). Thus, we made attempts to identify the receptor of dithranol by using aryl hydrocarbon receptor (AhR) knockout mice in order to substantiate our findings on the functional level. However, it appears that dithranol does not act via modulation of AhR, as we did not observe differences in the response of skin to dithranol comparing AhR knockout mice to controls. Certainly, future studies are indeed needed to identify dithranol’s yet unknown receptor. This all has now been worded in the Discussion.

5) The increase in epidermal thickening, the changes in keratin 16 and AMPs found in the mouse tail experiment with dithranol contradict the results found in human psoriasis (Figure 4).

We agree with the reviewer, the effects of dithranol in the mouse-tail test seem to be contradictory. However, this test has limitations since its usefulness only refers to which degree an agent or treatment increases orthokeratosis as a measure of keratinocyte differentiation-inducing activity (Figure 3—figure supplement 1), as we have demonstrated it for dithranol. On the other hand, as the reviewer observed, besides upregulation of keratinocyte markers dithranol increased certain psoriasis-associated AMPs such as *Defb3* and *S100a8* and *S100a9* but decreased the pro-psoriatic antimicrobial peptide *Camp/LL37* as well as *Cxcl5*, a chemotactic factor for neutrophils (Figure 3—figure supplement 2). We now have mentioned this in the Discussion, together with a couple of new references and stressed the limitations of the mouse tail-test (third paragraph).

6) IL-36 cytokines are mainly expressed by keratinocytes (Wenzel et al., J Invest Dermatol 2015), but the authors describe a decrease at week 2 to 3. If dithranol targets primarily keratinocytes and inhibits IL-36, the effect should be visible at earlier time points.

We thank the reviewer for providing the reference of Wenzel et al. with regard to IL-36 expression in keratinocytes. Although we already saw a reduction of keratinocyte-related genes at day 6, a lot of these genes were only normalized at week 2-3, accompanied with a significantly diminished mRNA expression of pro-psoriatic IL-1 family members (*IL36A*, *IL36G* and *IL36RN*) (Supplementary file 1 and Figure 4). At day 6, PASI had decreased by 33%, but at week 2-3 we saw a reduction of 58%. Most likely, dithranol’s effect on members of the IL-36 family might be linked to its strong clinical effect at week 2-3. The therapeutic importance of reduction of IL-1 family members in human skin was substantiated by results generated in the keratinocyte-based c-Jun/JunB mouse psoriasis model, where IL-36 family genes were significantly reduced in dithranol-treated psoriatic c-Jun/JunB lesions compared to controls. We have added a respective sentence on IL-36 in the first paragraph of the Discussion.

Reviewer #3:[…] I have the following comments:1) Although the data in the manuscript clearly shows that the early mode of action of dithranol in psoriasis treatment is on keratinocyte-derived molecules and subsequently neutrophil infiltration to the skin the manuscript does not reveal the molecular mechanism behind this. Does dithranol use a membrane bound receptor, an intracellular receptor or does it inhibit specific transcription factors (e.g. NF-κB, STAT or IkBzeta)? Have the authors investigated this or is there anything in the literature about this. The authors should discuss this more in the manuscript.

We agree with the reviewer on that point and have discussed the limitations of our work now in more detail (Discussion, ninth paragraph). What indeed our work does not answer, is how dithranol exactly acts at the molecular level, which we also mention in the ninth paragraph of the Discussion, paragraph 1. Little is known in the literature about dithranol's potential receptor that remains yet to be determined. Please see also our response in this regard to comment 4 of reviewer 2 about the aryl hydrocarbon receptor (AhR).

2) Surprisingly, dithranol does not have an anti-psoriatic effect in the imiquimod-induced psoriasis model. In contrast, dithranol has an anti-psoriatic effect in the c-Jun/JunB mouse model. Were the mouse strains used in the two models the same? Were the mice similar in age and did the authors use females or males and was this the same in the two models? The authors write that dithranol rather worsened the psoriatic lesions in the IMQ model. And interestingly the authors write that "Dithranol treatment of healthy murine skin led to similar effects". Could this be the same effect as has been described in perilesional skin? Is the inflammatory effect of dithranol on perilesional skin known? If so the authors could examine if this is also what is seen in skin of IMQ-treated mice. These issues could be discussed in more detail.

As stated in the manuscript (see subsection “Animals”), we used 6-9 week-old mice with different strains. BALB/c mice are commonly used for the Imiquimod model, as we did; the c-Jun/JunB knockout mice of our study have a mixed background, as described by Zenz et al., 2005. However, it is highly unlikely that the different mouse strains played any role in the outcome of the experiments since the two psoriasis models are per se completely different. Immunological aspects are important in the imiquimod model and molecular alterations dominate the c-Jun/JunB model. Concerning the effect of dithranol on healthy murine skin we have added a couple of sentences on dithranol’s irritant effect that may be rather a side effect without functional relevance in human psoriasis (Discussion, fourth paragraph) but crucial for dithranol's action in alopecia areata (see response to the comment 3 of reviewer #1 above). We now have added a couple of respective sentences on alopecia areata in the Discussion.

3) In the psoriasis patients treated with dithranol biopsies were taken from lesional skin over time. Did the authors take biopsies from the same "target" lesion throughout the study?

We have now precisely outlined how marker lesions were defined, scored, treated and biopsy-sampled in the Materials and methods subsection under the new heading “Marker lesions and scoring”. This has also been illustrated in Figure 1A.

4) In Table 1 the authors should in the legend include that the numbers in the table are based on data from 15 psoriatic patients.

We thank the reviewer for mentioning this and have changed the heading of Table 1 accordingly.